# Constructing Adversarial Examples for Vertical Federated Learning: Optimal Client Corruption through Multi-Armed Bandit

**Duanyi, Yao**
HKUST
dyao@connect.ust.hk

**Songze, Li**
Southeast University
songzeli@seu.edu.cn

**Ye, Xue**
Shenzhen Research Institute of Big Data, CUHK(SZ)
xueye@cuhk.edu.cn

**Jin, Liu**
HKUST(GZ)
jliu577@connect.hkust-gz.edu.cn

## Abstract

Vertical federated learning (VFL), where each participating client holds a subset of data features, has found numerous applications in finance, healthcare, and IoT systems. However, adversarial attacks, particularly through the injection of adversarial examples (AEs), pose serious challenges to the security of VFL models. In this paper, we investigate such vulnerabilities through developing a novel attack to disrupt the VFL inference process, under a practical scenario where the adversary is able to *adaptively corrupt a subset of clients*. We formulate the problem of finding optimal attack strategies as an online optimization problem, which is decomposed into an inner problem of adversarial example generation (AEG) and an outer problem of corruption pattern selection (CPS). Specifically, we establish the equivalence between the formulated CPS problem and a multi-armed bandit (MAB) problem, and propose the Thompson sampling with Empirical maximum reward (E-TS) algorithm for the adversary to efficiently identify the optimal subset of clients for corruption. The key idea of E-TS is to introduce an estimation of the expected maximum reward for each arm, which helps to specify a small set of *competitive arms*, on which the exploration for the optimal arm is performed. This significantly reduces the exploration space, which otherwise can quickly become prohibitively large as the number of clients increases. We analytically characterize the regret bound of E-TS, and empirically demonstrate its capability of efficiently revealing the optimal corruption pattern with the highest attack success rate, under various datasets of popular VFL tasks.

## 1 Introduction

Federated learning (FL) Li et al. (2020) is a distributed learning paradigm that enables multiple clients to collaboratively train and utilize a machine learning model without sharing their data. Conventionally, most FL research considers the Horizontal FL (HFL) setting, where clients hold different data samples with the same feature space. In contrast, vertical FL (VFL) tackles the secnarios where clients have identical samples but disjoint feature spaces. A typical VFL model comprises a top model maintained by a server and multiple bottom models, one at each participating client. During the inference process, each client computes the local embedding of data features using its bottom model and uploads it to the server through a communication channel for further prediction. Due to its advantage of incorporating attributes from diverse information sources, VFL has found promising applications in healthcare systems Poirot et al. (2019), e-commerce platforms Mammen (2021), and financial systems Liu et al. (2022). VFL inference has also been applied to the Internet of Things (IoT) scenarios (also known as collaborative inference Liu et al.; Ko et al. (2018)), where sensor data with distinct features are aggregated by a fusion center for further processing. A recent example is to utilize multi-modal image data from sensors for remote sensing image classification Shi et al. (2022).

Despite its widespread applications, ML models have been shown vulnerable to adversarial examples (AEs) Goodfellow et al. (2014), which are modified inputs designed to cause model misclassification during the inference. Constructing AEs in the VFL setting presents unique challenges compared to the conventional ML setting. Specifically, we consider a third-party adversary who can access, replay, and manipulate messages on the communication channel between a client and the server. (For simplicity, we use client $x$ to denote the communication channel between client $x$ and the server throughout the paper). However, it can only corrupt a subset of clients due to resource constraints, like computational and network bandwidth Lesi et al. (2020); Li and Tang (2018); Wu et al. (2018). Also, the server's top model and the other uncorrupted clients' embeddings and models are unknown to the adversary. Under this setting, the adversary aims to generate AEs by adding manipulated perturbations to embeddings in the corrupted clients, such that the attack success rate (ASR) over a sequence of test samples is maximized. Prior works have proposed methods to generate AEs for VFL inference, for a *fixed corruption pattern* (i.e., the set of corrupted clients remains fixed throughout the attack). In Pang et al. (2022), a finite difference method was proposed to generate adversarial dominating inputs, by perturbing the features of a fixed corrupted client, to control the inference result, regardless of the feature values of other clients; another work Qiu et al. (2022) employed zeroth-order optimization (ZOO) to find the optimal perturbation on the uploaded embedding of a malicious client. Meanwhile, these attacks also make assumptions on certain prior knowledge at the adversary, e.g., the adversary can obtain a subset of complete test samples in advance.

In this paper, we consider an adversary who lacks prior knowledge on test data or VFL models, but can *adaptively adjust its corruption pattern* based on the effectiveness of the previous attacks, subject to a maximum number of clients that can be corrupted. For a VFL inference process of $T$ rounds, we formulate the attack as an online optimization problem, over $T$ corruption patterns, one for each inference round, and the embedding perturbations for the test samples in each round. To solve the problem, we first decompose it into two sub-problems: the inner *adversarial examples generation (AEG)* problem and the outer *corruption pattern selection (CPS)* problem. For the AEG problem with a fixed corruption pattern, we apply the natural evolution strategy (NES) to estimate the gradient for perturbation optimization. For the outer CPS problem, we establish its equivalence with arm selection in a multi-armed bandit (MAB) problem, with the reward being the optimal ASR obtained from the AEG problem. Given the unique challenge that the total number of arms scale combinatorially with the number of clients, we propose a novel method named Thompson sampling with empirical maximum reward (E-TS), enabling the adversary to efficiently identify the optimal corruption pattern. The key idea is to limit the exploration within the competitive set, which is defined using the expected maximum reward of each arm. Compared with plain Thompson sampling (TS) for the MAB problem Agrawal and Goyal (2012), E-TS additionally maintains the empirical maximum reward for each arm, which are utilized to estimate the underlying competitive arms, within which TS is executed to select the corruption pattern.

We theoretically characterize a regret bound of $(N - D)\mathcal{O}(1) + D\mathcal{O}(\log(T))$ for the proposed E-TS algorithm, where $N$ is the number of arms and $D$ is the number of competitive arms. This demonstrates the advantage of E-TS over the plain TS, especially for a small number of competitive arms. We also empirically evaluate the performance of the proposed attack on datasets with four major types of VFL tasks. In all experiments, the proposed attack uniformly dominates all baselines with fastest convergence to the optimal corruption pattern with the highest ASR. For the proposed attack, we further conduct extensive experiments to evaluate its effectiveness under various combinations of system parameters and the design parameter, and common defense strategies against AEs.

## 2   PRELIMINARIES

**VFL inference.** A VFL system consists of a central server and $M$ clients. Each client $m \in [M]$ possesses a subset of disjoint features $\boldsymbol{x}_m$ and a corresponding bottom model $\boldsymbol{f}_m$, where $[M]$ denotes the set $\{1, \ldots, M\}$. The central server maintains a top model $\boldsymbol{f}_0$. Given a test sample $\boldsymbol{X} = [\boldsymbol{x}_1, \ldots, \boldsymbol{x}_M]$, VFL inference is carried out in two stages. First, each client $m$ computes a local embedding $\boldsymbol{h}_m = \boldsymbol{f}_m(\boldsymbol{x}_m)$ using its bottom model, and uploads it to the server through a communication channel for querying a label. In the second stage, the server aggregates the embeddings from all clients and computes the predicted result $\boldsymbol{p} = \boldsymbol{f}_0([\boldsymbol{h}_1, \ldots, \boldsymbol{h}_M]) \in \mathbb{R}^c$, which is a probability vector over $c$ classes. The server broadcasts $\boldsymbol{p}$ to all the clients, who then obtain the predicted label $\hat{y}(\boldsymbol{p})$, where $\hat{y}(\cdot)$ returns the class with the highest probability. To enhance robustness

against system impairments, such as dropouts in embedding transmission, the system permits repeated queries for the same test sample, with a maximum limit of $Q$ times. The inference process operates in an online mode, such that test samples are received continuously in a streaming fashion.

**Multi-armed bandit.** A multi-armed bandit (MAB) problem consists of $N$ arms. Each arm $k \in [N]$ corresponds to a random reward following an unknown distribution with mean $\mu_k$. The bandit is played for $T$ rounds. In each round $t \in [T]$, one of the $N$ arms, denoted by $k(t)$, is pulled. The pulled arm yields a random reward $r_{k(t)}(t)$ supported in the range $[0, 1]$, which is i.i.d. from repeated plays of the same arm and observed by the player. The player must decide which arm to pull at each round $t$, i.e., $k(t)$, based on the rewards in previous rounds, to maximize the expected cumulative reward at round $T$, expressed as $\mathbb{E}[\sum_{t=1}^{T} \mu_{k(t)}]$, where $\mu_{k(t)} = \mathbb{E}_t[r_{k(t)}(t)]$. Assuming there exists an optimal arm with the highest mean reward $\mu^*$, the problem is equivalent to minimizing the expected regret $\mathbb{E}[R(T)]$, which is defined as follows:

$$\mathbb{E}[R(T)] = \mathbb{E}\left[\sum_{t=1}^{T}(\mu^* - \mu_{k(t)})\right] = \mathbb{E}\left[\sum_{k=1}^{N} n_k(T)\Delta_k\right], \tag{1}$$

where $n_k(T)$ denotes the number of times pulling arm $k$ in $T$ rounds, and $\Delta_k = \mu^* - \mu_k$ denotes the mean reward gap between the optimal arm and arm $k$.

## 3 THREAT MODEL

**Goal of the adversary.** We consider two types of attacks: targeted attack and untargeted attack. For the targeted attack with some target label $y_v$, the adversary aims to corrupt the samples whose original prediction is not $y_v$, making the top model output $\hat{y} = y_v$. For instance in a lending application, the adversary might set $y_v$ to "lending" to secure a loan to an unqualified customer. For the untargeted attack with some label $y_u$, the adversary would like to corrupt the samples whose original prediction is $y_u$, making the top model output $\hat{y} \neq y_u$. Note that the conventional untargeted attack Mahmood et al. (2021) is a special case of the one considered here, when setting $y_u$ as the true label of the attacked samples.

**Metric.** The attack's effectiveness is measured by *attack success rate* (ASR), which is defined as $ASR_v^s = \frac{\sum_{i=1}^{s} \mathbb{1}(\hat{y}(\boldsymbol{p}_i) = y_v)}{s}$ and $ASR_u^s = \frac{\sum_{i=1}^{s} \mathbb{1}(\hat{y}(\boldsymbol{p}_i) \neq y_u)}{s}$ for targeted and untargeted attack, respectively, where $s$ is the number of samples to be attacked, $\boldsymbol{p}_i$ is the probability vector of test sample $i$, and $\mathbb{1}(\cdot)$ is the indicator function.

**Capability of the adversary.** We consider an adversary as a third party in VFL inference, who can access, replay, and manipulate messages on the communication channel between two endpoints. This scenario stems from a man-in-the-middle (MITM) attack Conti et al. (2016); Wang et al. (2020), e.g., Mallory can open letters sent from Bob to Alice and change or resend their contents before handing over the letter to Alice. In VFL inference, a communication channel is established between each client and the server, through which embeddings and predictions are exchanged. The adversary can choose to corrupt any specific channel, e.g., client 1 (for simplicity, we use client $x$ to denote the communication channel between client $x$ and the server). However, due to resource constraints like computational power and network bandwidth (see, e.g., Lesi et al. (2020); Wang et al. (2016); Wu et al. (2018)), the adversary can corrupt at most $C \leq M$ clients. Formally, for a test sample $i$, the adversary can perturb the embeddings of up to $C$ clients, denoted as $\boldsymbol{h}_{i,a}$ with $|\boldsymbol{h}_{i,a}| \leq C$, to obtain $\tilde{\boldsymbol{h}}_{i,a}$ such that $\|\tilde{\boldsymbol{h}}_{i,a} - \boldsymbol{h}_{i,a}\|_\infty \leq \beta(ub_i - lb_i)$, where $ub_i$ and $lb_i$ represent the maximum and minimum values of the elements in $\boldsymbol{h}_{i,a}$ respectively, and $\beta \in [0, 1]$ is the perturbation budget of some simple magnitude-based anomaly detector.

**Adaptive corruption.** In the context of online inference, we focus on a class of powerful adversaries capable of *adaptively* adjusting their corruption patterns. In each attack round, the adversary perturbs the embeddings in the corrupted clients for a batch of test samples. In subsequent attack rounds, the sets of corrupted clients can be adjusted subject to the constraint $C$, exploiting feedbacks on attack performance from previous rounds.

## 4 PROBLEM DEFINITION

The attack proceeds in $T$ rounds. In each attack round $t \in [T]$, the adversary seeks to perturb a batch of $B^t$ test samples following a corruption pattern $\mathcal{C}^t = \{a_1, \ldots, a_C\}$, where $a_j, j \in [C]$, denotes the

index of the corrupted client. More precisely, given the embeddings of a test sample $i \in [B^t]$, denoted as $\boldsymbol{h}_i^t = [\boldsymbol{h}_{i,1}^t, \ldots, \boldsymbol{h}_{i,M}^t]$, where $\boldsymbol{h}_{i,m}^t, m \in [M]$, represents the embedding vector of client $m$, we partition $\boldsymbol{h}_i^t$ into the adversarial part $\boldsymbol{h}_{i,a}^t = [\boldsymbol{h}_{i,a_1}^t, \ldots, \boldsymbol{h}_{i,a_C}^t]$, and the benign part $\boldsymbol{h}_{i,b}^t$, according to $\mathcal{C}^t$. The adversary crafts a perturbation $\boldsymbol{\eta}_i^t = [\boldsymbol{\eta}_{i,a_1}^t, \ldots, \boldsymbol{\eta}_{i,a_C}^t]$ with $\|\boldsymbol{\eta}_i^t\|_\infty \leq \beta(ub_i - lb_i)$, and adds it to $\boldsymbol{h}_{i,a}^t$ to obtain an adversarial embedding $\tilde{\boldsymbol{h}}_{i,a}^t = \boldsymbol{h}_{i,a}^t + \boldsymbol{\eta}_i^t$, before submitting it to the server. Upon receiving $\tilde{\boldsymbol{h}}_{i,a}^t$ and $\boldsymbol{h}_{i,b}^t$, the server returns the prediction $\boldsymbol{f}_0(\tilde{\boldsymbol{h}}_{i,a}^t; \boldsymbol{h}_{i,b}^t)$ to all clients. After collecting all predictions of $B^t$ adversarial embeddings, the adversary computes the ASR, i.e., $A(\{\boldsymbol{\eta}_i^t\}_{i=1}^{B^t}, \mathcal{C}^t; B^t) = \frac{\sum_{i=1}^{B^t} \mathbb{1}(\hat{y}(\boldsymbol{f}_0(\tilde{\boldsymbol{h}}_{i,a}^t; \boldsymbol{h}_{i,b}^t)) = y_v)}{B^t}$ for the targeted attack with target label $y_v$, or $A(\{\boldsymbol{\eta}_i^t\}_{i=1}^{B^t}, \mathcal{C}^t; B^t) = \frac{\sum_{i=1}^{B^t} \mathbb{1}(\hat{y}(\boldsymbol{f}_0(\tilde{\boldsymbol{h}}_{i,a}^t; \boldsymbol{h}_{i,b}^t)) \neq y_u)}{B^t}$ for the untargeted attack with label $y_u$.

The adversary aims to find the optimal set of corruption patterns $\{\mathcal{C}^t\}_{t=1}^T$, and the optimal set of perturbations $\{\{\boldsymbol{\eta}_i^t\}_{i=1}^{B^t}\}_{t=1}^T$ for each sample $i \in [B^t]$ in attack round $t \in [T]$, thus maximizing the expected cumulative ASR over $T$ attack rounds. We formulate this attack as an online optimization problem in (2). Note that the expectation $\mathbb{E}_t$ is taken over the randomness with the $t$-th attack round and the expectation $\mathbb{E}$ is taking over the randomness of all $T$ rounds.

$$\max_{\{\mathcal{C}^t\}_{t=1}^T} \frac{\mathbb{E}\left[\sum_{t=1}^T \mathbb{E}_t\left[\max_{\{\boldsymbol{\eta}_i^t\}_{i=1}^{B^t}} A(\{\boldsymbol{\eta}_i^t\}_{i=1}^{B^t}, \mathcal{C}^t; B^t)\right]\right]}{\sum_{t=1}^T B^t}$$
$$\text{s.t. } |\mathcal{C}^t| = C, \ \|\boldsymbol{\eta}_i^t\|_\infty \leq \beta(ub_i - lb_i), \ \forall t \in [T]. \tag{2}$$

## 5 METHODOLOGY

To solve Problem (2), we decompose the above problem into an inner problem of *adversarial example generation* (AEG) and an outer problem of *corruption pattern selection* (CPS). We first specify the inner problem of AEG. At each round $t$, $t \in [T]$, with a fixed corruption pattern $\mathcal{C}^t$, for each test sample $i \in [B^t]$, the adversary intends to find the optimal perturbation $\boldsymbol{\eta}_i^t$ that minimizes some loss function, as shown in (3). We consider the loss function $L(\boldsymbol{\eta}_i^t; \mathcal{C}^t) = l(\boldsymbol{f}_0(\tilde{\boldsymbol{h}}_{i,a}^t; \boldsymbol{h}_{i,b}^t), y_v)$ for the targeted attack with target label $y_v$, and $L(\boldsymbol{\eta}_i^t; \mathcal{C}^t) = -l(\boldsymbol{f}_0(\tilde{\boldsymbol{h}}_{i,a}^t; \boldsymbol{h}_{i,b}^t), y_u)$ for the untargeted attack with label $y_u$, where $l(\cdot)$ denotes the loss metric, such as cross-entropy or margin loss.

$$\text{Inner problem (AEG):} \quad \min_{\boldsymbol{\eta}_i^t} L(\boldsymbol{\eta}_i^t; \mathcal{C}^t), \quad \text{s.t. } \|\boldsymbol{\eta}_i^t\|_\infty \leq \beta(ub_i - lb_i), \forall i \in [B^t]. \tag{3}$$

Then, we obtain the ASR of $B^t$ test samples, i.e., $A^*(\mathcal{C}^t; B^t) = A(\{\boldsymbol{\eta}_i^{t*}\}_{i=1}^{B^t}, \mathcal{C}^t; B^t)$, obtained using optimal perturbations $\boldsymbol{\eta}_i^{t*}, i \in [B^t]$, from solving the problem AEG. As such, the outer problem of CPS can be cast into

$$\text{Outer problem (CPS):} \quad \min_{\{\mathcal{C}^t\}_{t=1}^T} \frac{\mathbb{E}\left[\sum_{t=1}^T (\alpha^* - \mathbb{E}_t\left[A^*(\mathcal{C}^t; B^t)\right]\right]}{\sum_{t=1}^T B^t}$$
$$\text{s.t. } |\mathcal{C}^t| = C, \ \forall t \in [T], \tag{4}$$

where $\alpha^*$ is any positive constant. The inherent randomness of $A^*(\mathcal{C}^t; B^t)$ for a fixed $\mathcal{C}^t$ arises from the random test samples and the random noises in the AE generation process.

### 5.1 AE GENERATION BY SOLVING THE AEG PROBLEM

To address the box-constraint inner AEG problem (3), one might initially consider employing the projected gradient descent (PGD) method Madry et al. (2017). However, in our setting, the adversary can only access the value of the loss function and cannot directly obtain the gradient, thus necessitating the use of ZOO methods. The ZOO method iteratively seeks for the optimal variable. Each iteration typically commences with an estimation of the current variable's gradient, followed by a gradient descent-based variable update. NES Ilyas et al. (2018), a type of ZOO method, not only estimates the gradient but also requires fewer queries than conventional finite-difference methods. NES is thus

---

**Algorithm 1** E-TS for CPS

---

1: **Initialization:** $\forall k \in [N], \hat{\mu}_k = 0, \hat{\sigma}_k = 1, n_k = 0, r_k^{\max} = 0, \hat{\varphi}_k = 0$.
2: **for** $t = 1, 2, \ldots, T$ **do**
3:     **if** $t > t_0$ **then**
4:         Select fully explored arms to construct the set $\mathcal{S}_t = \{k \in [N] : n_k \geq \frac{(t-1)}{N}\}$.
5:         Select the empirical best arm $k^{emp}(t) = \max_{k \in \mathcal{S}_t} \hat{\mu}_k$.
6:         Initialize $\mathcal{E}^t = \emptyset$, add arms $k \in [N]$ which satisfy $\hat{\varphi}_k \geq \hat{\mu}_{k^{emp}(t)}$ to $\mathcal{E}^t$.
7:     **else**
8:         Initialize set $\mathcal{E}^t = [N]$.
9:     **end if**
10:    $\forall k \in \mathcal{E}^t$: Sample $\theta_k \sim \mathcal{N}(\hat{\mu}_k, \hat{\sigma}_k)$.
11:    Choose the arm $k(t) = \arg\max_k \theta_k$ and decide the corruption pattern $\mathcal{C}^t = k(t)$.
12:    Sample batch data $[B^t]$, play the arm $k(t)$ as the corruption pattern in Algorithm 2 and observe the reward $r_{k(t)}(t)$ from the attack result for the corrupted embedding $\boldsymbol{h}_{i,a}^t = [\boldsymbol{h}_{i,a_1}^t, \ldots, \boldsymbol{h}_{i,a_C}^t], \forall i \in [B^t]$.
13:    Update $n_{k(t)} = n_{k(t)} + 1$, $\hat{\mu}_{k(t)} = \frac{\hat{\mu}_{k(t)}(n_{k(t)}-1)+r_{k(t)}(t)}{n_{k(t)}}$, $\hat{\sigma}_{k(t)} = \frac{1}{n_{k(t)}+1}$, $r_{k(t)}^{\max} = \max\{r_{k(t)}^{\max}, r_{k(t)}(t)\}$, $\hat{\varphi}_{k(t)} = \frac{\hat{\varphi}_{k(t)}(n_{k(t)}-1)+r_{k(t)}^{\max}}{n_{k(t)}}$.
14: **end for**
15: Output $\{k(1), \ldots, k(T)\}$

---

especially well-suited for addressing the AEG problem (3) in the VFL setting, where query times are inherently limited. In the process of AE generation using NES, the adversary samples $n$ Gaussian noises $\boldsymbol{\delta}_j \sim \mathcal{N}(\boldsymbol{0}, \boldsymbol{I})$, $j \in [n]$, and adds them to the current variable $\boldsymbol{\eta}_i^t$, with some scaling parameter $\sigma > 0$. Then, the gradient estimation is given by

$$\nabla_{\boldsymbol{\eta}_i^t} L(\boldsymbol{\eta}_i^t; \mathcal{C}^t) \approx \frac{1}{\sigma n} \sum_{j=1}^n \boldsymbol{\delta}_j L\left(\boldsymbol{\eta}_i^t + \sigma \boldsymbol{\delta}_j; \mathcal{C}^t\right). \tag{5}$$

After obtaining the gradient estimates, the adversary can update $\boldsymbol{\eta}_i^t$ in a PGD manner. The details of the AE generation process are provided in Algorithm 2 in Appendix A. Note that the number of queries on each test sample is limited to $Q$, therefore, the adversary can update the drafted perturbation at most $\lfloor \frac{Q}{n} \rfloor$ times for each sample.

### 5.2 THOMPSON SAMPLING WITH THE EMPIRICAL MAXIMUM REWARD FOR SOLVING THE CPS PROBLEM

To solve the CPS problem, we make a key observation that *the outer problem in (4) can be cast as an MAB problem*. Specifically, picking $C$ out of total $M$ clients to corrupt results in $N = \binom{M}{C}$ possible corruption patterns, which are defined as $N$ arms in the MAB problem. That is to say, there is a bijection between the set of $N$ arms and the optimization space of $\mathcal{C}^t$. Therefore, we can transform optimization variables $\{\mathcal{C}^t\}_{t=1}^T$ in (4) into the selected arms at $t$ round, i.e., $\{k(t)\}_{t=1}^T$. At round $t$, pulling an arm $k(t)$ returns the reward $r_{k(t)}(t)$ as the ASR, i.e., $r_{k(t)}(t) = A^*(\mathcal{C}^t; B^t) \in [0, 1]$. We define the mean of the reward for arm $k(t)$ as $\mathbb{E}_t[r_{k(t)}(t)] = \mu_{k(t)} = \mathbb{E}_t[A^*(\mathcal{C}^t; B^t)]$. Without loss of generality, we assign the best arm the arm 1 with fixed positive mean $\mu_1 > 0$, which can be considered as the positive value $\alpha^*$ in (4). Finally, the CPS problem in (4) is transformed into an MAB problem, i.e., $\min_{\{(k(t))\}_{t=1}^T} \mathbb{E}\left[\sum_{t=1}^T (\mu_1 - \mu_{k(t)})\right]$.

**E-TS algorithm.** In our context, the adversary could face a significant challenge as the exploration space $N$ can become prohibitively large when engaging with hundreds of clients, which could result in a steep accumulation of regret. To mitigate the issue from extensive exploration, we first introduce the following definition of the competitive arm.

**Definition 1 (Competitive arm).** An arm $k$ is described as a competitive arm when the expected maximum reward is larger than the best arm's mean, i.e., $\tilde{\Delta}_{k,1} = \frac{\sum_{t=1}^T \mathbb{E}[r_k^{\max}(t)]}{T} - \mu_1 \geq 0$, where $r_k^{\max}(t) = \max_{\tau \in [t]}\{r_k(\tau)\}$. Otherwise, it is a non-competitive arm.

Based on the above definition, we propose *Thompson sampling with Empirical maximum reward* (E-TS) algorithm. The basic idea of E-TS is to restrict the exploration space within the set of competitive arms to reduce accumulated regret. However, the ground-truth competitive arms cannot be accessed a priori. Therefore, we propose to construct an *empirical competitive set* $\mathcal{E}^t$ with estimated competitive arms at each round $t$ and restrict exploration within it. Estimating the competitive arms requires calculating the *empirical best arm* and *empirical maximum reward* defined as follows.

**Definition 2 (Empirical best arm and empirical maximum reward).** An arm $k$ is selected as the empirical best arm $k^{emp}(t)$ at round $t$, when $k = \arg\max_{k \in \mathcal{S}^t} \hat{\mu}_k(t)$, where $\hat{\mu}_k(t)$ is the estimated mean of arm $k$'s reward at round $t$, $\mathcal{S}^t = \{k \in [N] : n_k(t) \geq \frac{(t-1)}{N}\}$, and $n_k(t)$ denotes the number of times pulling arm $k$ in $t$ rounds. An arm $k$'s empirical maximum reward $\hat{\varphi}_k(t)$ is computed by: $\hat{\varphi}_k(t) := \frac{\sum_{\tau=1}^t r_k^{\max}(\tau) \mathbb{1}(k(\tau)=k)}{n_k(t)}$.

Based on Definitions 1 and 2, we are now able to present the key components of the E-TS algorithm. E-TS consists of two steps: first, for constructing an empirical competitive set $\mathcal{E}^t$ at round $t$, E-TS estimates $\mu_1$ and $\frac{\sum_{t=1}^T \mathbb{E}[r_k^{\max}(t)]}{T}$ using the mean of empirical best arm $\hat{\mu}_{k^{emp}(t)}(t)$ and the empirical maximum reward $\hat{\varphi}_k(t)$, and obtains $\mathcal{E}^t = \{k \in [N] : \hat{\varphi}_k(t) - \hat{\mu}_{k^{emp}(t)}(t) \geq 0\}$. Second, while performing TS to explore each arm, E-TS adopts a Gaussian prior $\mathcal{N}(\hat{\mu}_k(t), \frac{1}{n_k(t)+1})$ to approximate the distribution of the reward, where $\hat{\mu}_k(t)$ is defined as $\hat{\mu}_k(t) := \frac{\sum_{\tau=1}^t r_k(\tau) \mathbb{1}(k(\tau)=k)}{n_k(t)}$. In addition to the above two steps, E-TS also involves $t_0$ warm-up rounds, in which it simply executes TS across all arms. These warm-up rounds are designed to facilitate a more accurate estimation of each arm's reward mean and expected maximum reward. The complete algorithm is presented in Algorithm 1.

*Remark* 1. Previous work Gupta et al. (2021) leverages the upper bound $s_{k,l}(r)$ of arm $k$'s reward conditioned on obtaining reward $r$ from pulling arm $l$ (i.e., $\mathbb{E}[r_k(t)|r_l(t) = r] \leq s_{k,l}(r)$) to reduce the exploration space, where $s_{k,l}(r)$ is a known constant. In contrast, the proposed E-TS algorithm does not require any prior information about reward upper bound, making it more practical.

## 6 REGRET ANALYSIS

In this section, we analyze the regret bound for the proposed E-TS algorithm. Prior to proof, we assume that each arm is pulled at least twice during the initial warm-up rounds. This assumption aligns with our analysis on the optimal choice of warm-up rounds detailed in Appendix C.6. Achieving this assumption is highly probable as the number of warm-up rounds increases asymptotically Agrawal and Goyal (2017). Additionally, an adversary can traverse all arms before implementing E-TS to ensure this prerequisite is met. To facilitate discussion, we first introduce two key lemmas. Then, we present the expected regret bound of E-TS algorithm in Theorem 1. We defer all proof details of the lemmas and the theorem in Appendix B.

**Lemma 1 (Expected pulling times of a non-competitive arm).** *Under the above assumption, for a non-competitive arm $k^{nc} \neq 1$ with $\tilde{\Delta}_{k^{nc},1} < 0$, the expected number of pulling times in $T$ rounds, i.e., $\mathbb{E}[n_{k^{nc}}(T)]$, is bounded by $\mathbb{E}[n_{k^{nc}}(T)] \leq \mathcal{O}(1)$.*

**Lemma 2 (Expected pulling times of a competitive but sub-optimal arm).** *Under the above assumption, the expected number of times pulling a competitive but sub-optimal arm $k^{sub}$ with $\tilde{\Delta}_{k^{sub},1} \geq 0$ in $T$ rounds is bounded as follows,*

$$\mathbb{E}[n_{k^{sub}}(T)] = \sum_{t=1}^T \Pr(k(t) = k^{sub}, n_1(t) \geq \frac{t}{N}) \leq \mathcal{O}(\log(T)).$$

**Theorem 1 (Upper bound on expected regret of E-TS).** *Let $D \leq N$ denote the number of competitive arms. Under the above assumption, the expected regret of the E-TS algorithm is upper bounded by $D\mathcal{O}(\log(T)) + (N-D)\mathcal{O}(1)$.*

*Proof sketch.* We first demonstrate that the probability that pulling the optimal arm is infrequent (i.e., $n_1(t) < \frac{(t-1)}{N}$) is bounded. Next, we categorize the sub-optimal arms into non-competitive arms and competitive but sub-optimal arms, and analyse their regret bound respectively. For a non-competitive arm $k^{nc}$, the probability of $k(t) = k^{nc}$ is bounded by the probability of selecting as the competitive arm, i.e., $\Pr(k^{nc} \in \mathcal{E}^t)$, which is further bounded as in Lemma 1. On the other hand, for a competitive

but sub-optimal arm $k^{sub}$, we further divide the analysis in two cases based on whether or not the optimal arm is included in $\mathcal{E}^t$. By combining the probability upper bounds in these two cases, we arrive at an upper bound on the probability of $k(t) = k^{sub}$ as in Lemma 2. $\qquad\square$

*Remark* 2. In comparison with plain TS, our proposed E-TS holds a significant advantage in terms of limiting the expected number of times pulling a non-competitive arm, which is reduced from $\mathcal{O}(\log(T))$ to $\mathcal{O}(1)$.

## 7 EXPERIMENTAL EVALUATIONS

### 7.1 SETUP

The proposed attack is implemented using the PyTorch framework Paszke et al. (2017), and all experiments are executed on a single machine equipped with four NVIDIA RTX 3090 GPUs. Each experiment is repeated for 10 trials, and the average values and their standard deviations are reported.

**Datasets.** We perform experiments on six datasets of distinct VFL tasks. 1) Tabular dataset: **Credit** Yeh and Lien (2009) and **Real-Sim** Chang and Lin (2011), where data features are equally partitioned across 6 and 10 clients, respectively; 2) Computer vision (CV) dataset: **FashionMNIST** Xiao et al. (2017) and **CIFAR-10** Krizhevsky et al. (2009), with features equally distributed across 7 and 8 clients, respectively; 3) Multi-view dataset: **Caltech-7** Li et al. (2022), which consists of 6 views, each held by a separate client; 4) Natural language dataset: **IMDB** Maas et al. (2011), where each complete movie review is partitioned among 6 clients, each possessing a subset of sentences. More details about the datasets and model structures are provided in Appendix C.1.

**Baselines.** We consider three baseline strategies for corruption pattern selection: 1) **Fixed corruption pattern**, where the adversary corrupts a fixed set of clients during the inference. For comparison, we consider two fixed corruption patterns where one is the underlying optimal pattern with the highest ASR, and another is randomly selected at the beginning of the attack; 2) **Random corruption (RC)**, where the adversary selects uniformly at random a set of $C$ clients to corrupt in each attack round; and 3) **Plain Thompson sampling (TS)**, where the adversary executes the plain TS to improve the corruption pattern selection.

**Experimental parameters setting.** The adversary can query the server for up to $Q = 2000$ times per test sample. The number of warm-up rounds in E-TS $t_0$ is set to 80 for FashionMNIST, CIFAR-10, and Caltech-7, 50 for Credit and Real-Sim, and 40 for IMDB. For the targeted attack, we set the target label to 7 for FashonMNIST and CIFAR-10, and 3 for Caltech-7. We measure the ASR over 30 test epochs, each comprising multiple attack rounds. In our ablation study, we adjust one parameter at a time, keeping the rest constant, with default settings of $C = 2$, $t_0 = 80$, $Q = 2000$, and $\beta = 0.3$.

### 7.2 RESULTS

We plot the ASR of targeted and untargeted attacks for different datasets in Figure 1. Note that the targeted and untargeted attacks are equivalent for Credit, Real-Sim, and IMDB with binary labels. We observe that uniformly across all datasets, the proposed E-TS method effectively attacks VFL models with an ASR of $38\% \sim 99\%$ for targeted attack and $41\% \sim 99\%$ for untargeted attack. For each attack, we observe a significant gap in ASR between the best and sub-optimal corruption patterns, demonstrating the significance of corruption pattern selection. The RC baseline exhibits a stable, yet sub-optimal ASR performance, as it does not leverage any information from historical ASRs. In sharp contrast, the performance of both TS and E-TS converge to that of the best corruption pattern. Notably, thanks to the estimation of empirical maximum reward, the E-TS algorithm efficiently narrows down the exploration space, achieving a much faster and more stable convergence than TS.

**Ablation study.** We evaluate the effects of system parameters, including corruption constraint $C$, query budget $Q$, and perturbation budget $\beta$, and the design parameter, the number of warm-up rounds $t_0$, on the performance of the proposed attack. Besides, we test the attack performance under a larger search space. As shown in Figure 3(a), ASR increases as more clients are corrupted, and E-TS consistently outperforms random corruption. It is illustrated in Figure 3(b) that it is critical to select the appropriate number of warm-up rounds $t_0$ at the beginning of E-TS. When $t_0$ is too small, i.e., $t_0 = 20$, it leads to an inaccurate estimate of the empirical competitive set which may exclude the best arm, causing E-TS to converge on a sub-optimal arm. However, if $t_0$ is too large, i.e., $t_0 = 200$ or $1000$, the advantage over plain TS diminishes. That is, one needs to optimize $t_0$ to find the optimal arm with the fastest speed. Figure 3(c) and (d) show that ASR generally increases with

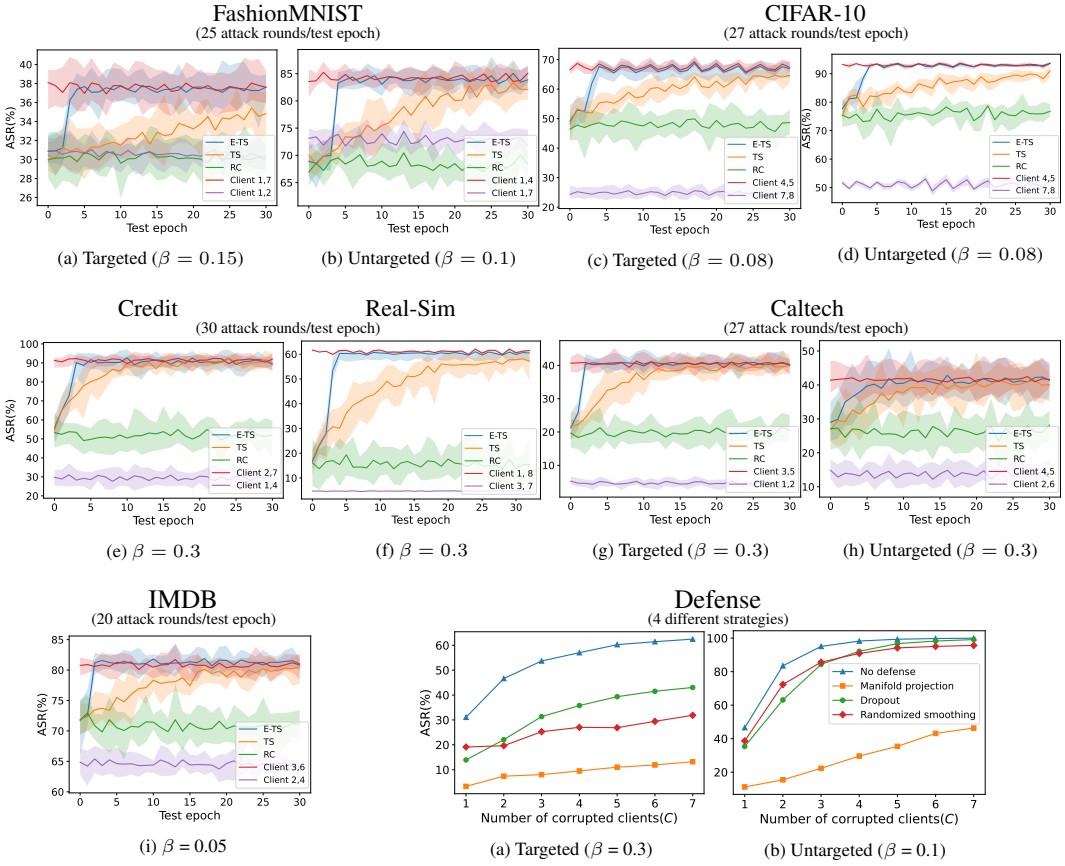

Figure 1: Attack performance on six datasets of distinct VFL tasks.

Figure 2: Attack performance on FashionMNIST under different defense strategies.

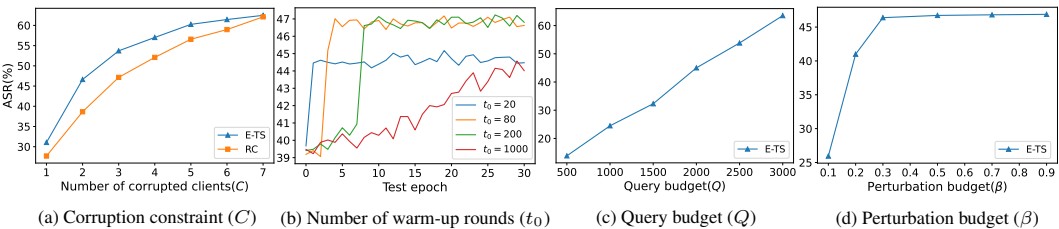

Figure 3: Targeted attack performance on FashionMNIST using different parameters.

larger $Q$ and $\beta$. Nevertheless, after reaching 0.3, increasing perturbation budget has negligible effect on improving ASR. Figure 4 shows that E-TS consistently outperforms baselines in larger exploration space, i.e., when there are $\binom{16}{2} = 120$, $\binom{16}{3} = 560$, $\binom{28}{2} = 378$, and $\binom{28}{3} = 3276$ choices. Notably, this performance gap between E-TS and TS becomes even more pronounced when the exploration space is expanded, demonstrating its effectiveness in handling larger exploration spaces. More experimental results of corrupting different numbers of clients on other datasets are provided in Appendix C.2. We also investigated the dynamics of arm selection and empirical competitive set in TS and E-TS (in Appendix C.3), minimum query budget and corruption channels to achieve 50% ASR (in Appendix C.4), the E-TS performance in large exploration spaces (in Appendix C.5), and the optimal choice on the warm-up round $t_0$ (in Appendix C.6).

**Defenses.** We further evaluate the effectiveness of the proposed attack under the following common defense strategies. **Randomized smoothing (Cohen et al. (2019)):** The main idea is to smooth out the decision boundary of a classifier, such that it's less sensitive to small perturbations in the input data. To construct a smooth VFL classifier, Gaussian noises are added to clients' embeddings, which are then processed by the top model to make a prediction. The final prediction is obtained by majority voting over 100 such trials; **Dropout (Qiu et al. (2022)):** A dropout layer is added after

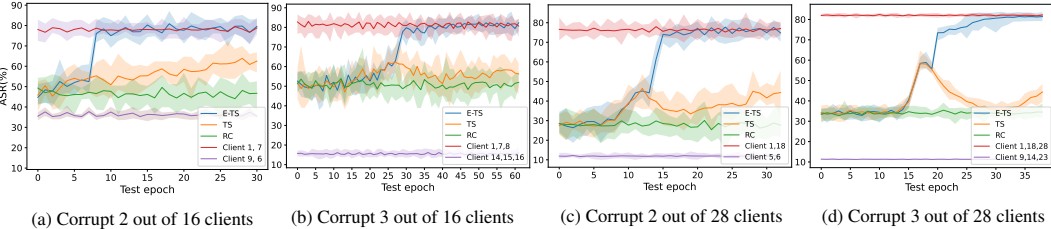

| (a) Corrupt 2 out of 16 clients | (b) Corrupt 3 out of 16 clients | (c) Corrupt 2 out of 28 clients | (d) Corrupt 3 out of 28 clients |

Figure 4: Targeted attack performance on FashionMNIST with larger search space

each activation layer in the server's top model to improve the robustness. Here, we set the dropout rate to 0.3; **Manifold projection (Meng and Chen (2017); Lindqvist et al. (2018)):** An autoencoder is incorporated into the model as a pre-processing step before the top model. During training, the autoencoder is trained using clean embeddings and designed to reconstruct the original embeddings. During inference, the clients' embeddings are first processed using the autoencoder before being passed to the top model for prediction.

As shown in Figure 2, for a targeted attack with $\beta = 0.3$, the ASR of the proposed attack reduces under all considered defenses; for an untargeted attack when $\beta = 0.1$, the ASRs experience marginal reductions under the randomized smoothing and dropout defenses, but significant drops of $35\% \sim 72\%$ in the ASR under manifold projection. The advantage of manifold projection can be attributed to the learning of the manifold structure and the transformation of adversarial embeddings into clean embeddings. Overall, while manifold projection exhibits the strongest capability in defending the proposed attack, it fails to completely eliminate all AEs.

## 8 RELATED WORK

**AE generation for ML models.** AE generation methods can be generally classified into two categories: white-box and black-box settings. While the former assumes the adversary knows full knowledge of model parameters and architectures, the latter assumes no prior knowledge of either the models or training data. Our work is concerned with a black-box setting, which is typically addressed using either transfer-based or query-based solutions. Transfer-based methods Papernot et al. (2016); Liu et al. (2016) generate AEs using a substitute model, which is trained either by querying the model's output or using a subset of training data. Query-based methods Bhagoji et al. (2018); Chen et al. (2017) optimize AEs utilizing gradient information, which is estimated through the queried outputs. One classical example is the ZOO attack Chen et al. (2017), which employs zeroth-order stochastic coordinate descent for gradient estimation.

**MAB algorithms.** Multiple classical algorithms, such as $\epsilon$-greedy Sutton and Barto (2018), Upper Confidence Bounds (UCB) Lai et al. (1985); Garivier and Cappé (2011), and Thompson sampling (TS) Agrawal and Goyal (2012), are proposed to solve the MAB problem. Recent advancements have proposed variants of MAB under different settings, leveraging additional information to minimize the exploration. These include correlated arm bandit Gupta et al. (2021), contextual bandit Singh et al. (2020); Chu et al. (2011), and combinatorial bandit Chen et al. (2013). However, their application in the context of adversarial attacks, particularly in VFL, remains largely unexplored.

## 9 CONCLUSION

We propose a novel attack, for an adversary who can adaptively corrupt a certain number of communication channels between a client and the server, to generate AEs for inference of VFL models. Specifically, we formulate the problem of adaptive AE generation as an online optimization problem, and decompose it into an adversarial example generation (AEG) problem and a corruption pattern selection (CPS) problem. We transform the CPS problem into an MAB problem, and propose a novel Thompson Sampling with Empirical maximum reward (E-TS) algorithm to find the optimal corruption pattern. We theoretically characterize the expected regret bound of E-TS, and perform extensive experiments on various VFL tasks to substantiate the effectiveness of our proposed attack.

### ACKNOWLEDGMENT

This work is in part supported by the National Nature Science Foundation of China (NSFC) Grant 62106057.

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

APPENDIX

## A  AE GENERATION ALGORITHM

The function (6) is used to estimate the gradient from the Natural evolution strategy (NES) Ilyas et al. (2018). The detailed method for zeroth-order AE generation in VFL is presented in Algorithm 2. In step 6, we use antithetic sampling to generate noise for efficiency.

$$
\nabla_{\boldsymbol{\eta}_i^t} L(\boldsymbol{\eta}_i^t, \mathcal{C}^t) \approx \frac{1}{\sigma n} \sum_{j=1}^n \boldsymbol{\delta}_j L\left(\boldsymbol{\eta}_i^t + \sigma \boldsymbol{\delta}_j, \mathcal{C}^t\right), \tag{6}
$$

---

**Algorithm 2** Zeroth-order AE generation in VFL

---

1: **Input:** Batch $[B^t]$, adversarial embedding $\boldsymbol{h}_{i,a}^t$, benign embedding $\boldsymbol{h}_{i,b}^t, i \in [B^t]$, corruption pattern $\mathcal{C}^t$, learning rate $lr$, the sample size of the Gaussion noise $n$, the perturbation budget $\beta$, query budget $Q$, and the embedding range $[lb_i, ub_i], i \in [B^t]$.
2: **Initialization:** $\boldsymbol{\eta}_{i,m}^t = \boldsymbol{0}, m \in [M]$, $\boldsymbol{\eta}_i^t = [\boldsymbol{\eta}_{i,a_1}^t, \dots, \boldsymbol{\eta}_{i,a_C}^t]$, counter $s = 0$.
3: **for** $i \in [B^t]$ **do**
4:     **for** $q \in [\frac{Q}{n}]$ **do**
5:         Clamp the perturbation to $\|\boldsymbol{\eta}_i^t\|_\infty \le \beta(ub_i - lb_i)$.
6:         Make a query to the server with adversarial embedding $\tilde{\boldsymbol{h}}_{i,a}^t = \boldsymbol{h}_{i,a}^t + \boldsymbol{\eta}_i^t$
7:         **if** the attack is not successful **then**
8:             Initiate $\frac{n}{2}$ noise vectors $\boldsymbol{\delta}_v \sim \mathcal{N}(0, I), v \in \{1, .., \frac{n}{2}\}$, another $\frac{n}{2}$ noise vectors are $\boldsymbol{\delta}_u = -\boldsymbol{\delta}_v, u \in \{\frac{n}{2}, ..., n\}$.
9:             Clamp the perturbation to $\|\boldsymbol{\eta}_i^t + \boldsymbol{\delta}_j\|_\infty \le \beta(ub_i - lb_i)$, where $j \in [n]$.
10:            Make $n$ queries to the server and estimate the gradient $\hat{\boldsymbol{G}}$ through function (6).
11:            Update the perturbation $\boldsymbol{\eta}_i^t = \boldsymbol{\eta}_i^t - lr * \hat{\boldsymbol{G}}$.
12:         **else**
13:            Break the loop, store $\boldsymbol{\eta}_i^t$ and $s = s + 1$.
14:         **end if**
15:     **end for**
16: **end for**
17: Clamp $\|\boldsymbol{\eta}_i^t\|_\infty \le \beta(ub_i - lb_i), i \in [B^t]$, return $\boldsymbol{\eta}_i^t$ and the attack success rate $\frac{s}{B^t}$.

---

## B  PROOFS IN SECTION REGRET ANALYSIS

In this section, we provide detailed proofs of lemmas and the theorem in **Section 6 Regret Analysis** of our paper. We initiate the proof procedure by establishing the definitions for two key events and three supporting facts, intended to streamline the proof process.

**Fact 1 (Hoeffding's inequality).** *Let $X_1, \dots, X_n$ be independent i.i.d. random variables bounded in $[a, b]$, then for any $\delta > 0$, we have*

$$
\Pr\left(\left|\frac{\sum_{i=1}^n X_i}{n} - \mathbb{E}(X_i)\right| \ge \delta\right) \le 2\exp\left(\frac{-2n\delta^2}{(b-a)^2}\right).
$$

**Fact 2 (Abramowitz And Stegun 1964).** *For a Gaussian distributed random variable $Z$ with mean $m$ and variance $\sigma^2$, for any $z$,*

$$
\frac{1}{4\sqrt{\pi}} \cdot e^{-7z^2/2} < \Pr(|Z - m| > z\sigma) \le \frac{1}{2} e^{-z^2/2}
$$

**Fact 3 (Concentration Bounds).** *Let $X_1, \dots, X_n$ be 0-1-valued random variables. Suppose that there are $0 \le \delta_i \le 1$, for $1 \le i \le n$, such that, for every set $S \subseteq [n]$, $\Pr[\wedge_{i \in S} X_i = 1] \le \prod_{i \in S} \delta_i$. Let $\delta = (1/n) \sum_{i=1}^n \delta_i$. Then, for any $\gamma$ such that $\delta \le \gamma \le 1$, we have $\Pr[\sum_{i=1}^n X_i \ge \gamma n] \le e^{-nD(\gamma\|\delta)}$, where $D(a\|b)$ is the cross entropy of $a$ and $b$.*

**Definition 3 (Events $E_1(t)$ and $E_2(t)$).** $E_1(t)$ is the event that the optimal arm 1 satisfies $n_1(t) < \frac{(t-1)}{N}, \forall t \in [T - t_0]$. $E_2(t)$ is the event that the optimal arm 1 is not identified in the empirical competitive set $\mathcal{E}^t$ at round $t$, $t > t_0$.

Based on the above facts and the definition, we then provide the following lemmas.

**Lemma 3.** *Let* $\gamma = \frac{N-1}{N}$, *and* $\delta = (N - 1)(\frac{1}{2}\exp(-\Delta_{\min}^2/16) + 2\exp(-\Delta_{\min}^2/4) - \frac{1}{2}\exp(-5\Delta_{\min}^2/16) + \exp\left(-\frac{(t_0-1)\Delta_{\min}^2}{2N}\right) + \exp(-\Delta_{\min}^2))$. *The probability of event $E_1(t)$ is upper bounded by* $\Pr(n_1(t) < \frac{t-1}{N}) \le \exp\left(-tD(\gamma\|\delta)\right)$.

*Proof.* Let $X_\tau = 0$ denote the optimal arm 1 is pulled at $\tau$ round, and $X_\tau = 1$ denotes that the best arm is not pulled. Considering the probability $\Pr(n_1(t) < \frac{t-1}{N}, t > t_0)$, we assume that each arm is pulled at least two times after the warm-up round $t_0$. Therefore, we can transform the probability $\Pr(n_1(t) < \frac{t}{N}, t > t_0)$ into $\Pr(\sum_{\tau=t_0+1}^t X_\tau > \frac{(N-1)}{N}t + \frac{2N+1}{N}) \le \Pr(\sum_{\tau=t_0+1}^t X_\tau > \frac{(N-1)}{N}t)$.

In our algorithm, for every set $S \subseteq [t - t_0]$, $\Pr(\wedge_{\tau \in S} X_\tau = 1) = \prod_{\tau \in S} \Pr(X_\tau = 1|\mathcal{F}_\tau)$, where $\mathcal{F}_\tau$ is the history of pulling the optimal arm 1 until round $\tau$. We first analyze the upper bound of the probability $\Pr(X_\tau = 1|\mathcal{F}_\tau)$, when $\tau \in [t - t_0]$.

From our algorithm, we can derive that $\Pr(X_\tau = 1|\mathcal{F}_\tau) \le \sum_{\ell \in [N]\backslash 1} \Pr(\theta_1(\tau) < \theta_\ell(\tau)|\mathcal{F}_\tau) + \sum_{\ell \in [N]\backslash 1} \Pr\left(\hat\varphi_1(\tau) < \hat\mu_\ell(\tau), n_\ell(t) \ge \frac{(\tau-1)}{N}\right)$, where $\ell$ is a sub-optimal arm. We then analyze the bound of probablity $\Pr(X_\tau = 1|\mathcal{F}_\tau)$ as follows:

$$
\begin{aligned}
&\sum_{\ell \in [N]\backslash 1} \Pr(\theta_1(\tau) < \theta_\ell(\tau)|\mathcal{F}_\tau) + \sum_{\ell \in [N]\backslash 1} \Pr\left(\hat\varphi_1(\tau) < \hat\mu_\ell(\tau), n_\ell(\tau) \ge \frac{(\tau-1)}{N}\right) \\
&\le \sum_{\ell \in [N]\backslash 1} \Pr\left(\left(\theta_1(\tau) < \mu_1 - \frac{\Delta_\ell}{2}\right)\bigcup\left(\theta_\ell(\tau) > \mu_1 - \frac{\Delta_\ell}{2}\right)\right) \\
&\quad + \sum_{\ell \in [N]\backslash 1} \Pr\left(\left(\hat\varphi_1(\tau) < \mu_1 - \frac{\Delta_{\min}}{2}\right)\bigcup\left(\hat\mu_\ell(\tau) > \mu_1 - \frac{\Delta_{\min}}{2}\right), n_\ell(\tau) \ge \frac{(\tau-1)}{N}\right) \\
&\overset{(a)}{\le} \sum_{\ell \in [N]\backslash 1} \Pr\left(\theta_1(\tau) < \mu_1 - \frac{\Delta_\ell}{2}\right) + \sum_{\ell \in [N]\backslash 1} \Pr\left(\theta_\ell(\tau) > \mu_\ell + \frac{\Delta_\ell}{2}\right) \\
&\quad + \sum_{\ell \in [N]\backslash 1} \Pr\left(\hat\varphi_1(\tau) < \frac{\sum_{t=1}^T \mathbb{E}[r_1^{\max}(t)]}{T} - \frac{\Delta_{min}}{2}\right) \\
&\quad + \sum_{\ell \in [N]\backslash 1} \Pr\left(\hat\mu_\ell(\tau) > \mu_\ell + \frac{\Delta_{min}}{2}, n_\ell(\tau) \ge \frac{(\tau-1)}{N}\right) \\
&\overset{(b)}{\le} (N - 1)((\frac{1}{2}\exp(-\Delta_{\min}^2/16) + 2\exp(-\Delta_{\min}^2/4) \\
&\quad - \frac{1}{2}\exp(-5\Delta_{\min}^2/16) + \exp\left(-\frac{(t_0-1)\Delta_{\min}^2}{2N}\right) + \exp(-\Delta_{\min}^2)),
\end{aligned}
\tag{7}
$$

where we have $(a)$ from the union bound. For inequality $(b)$, our objective is to delineate the upper bounds of to derive the upper bound of $\Pr\left(\theta_1(\tau) < \mu_1 - \frac{\Delta_\ell}{2}\right)$ and $\Pr\left(\theta_\ell(\tau) > \mu_\ell + \frac{\Delta_\ell}{2}\right)$. To achieve this, we invert our approach to discuss the lower bounds of $\Pr\left(\theta_1(\tau) \ge \mu_1 - \frac{\Delta_\ell}{2}\right)$ and $\Pr\left(\theta_\ell(\tau) \le \mu_\ell + \frac{\Delta_\ell}{2}\right)$. We first focus on the probability $\Pr\left(\theta_1(\tau) \ge \mu_1 - \frac{\Delta_\ell}{2}\right)$:

$$\Pr\left(\theta_1(\tau) \geq \mu_1 - \frac{\Delta_\ell}{2}\right) \geq \Pr\left(\theta_1(\tau) \geq \hat{\mu}_1(\tau) - \frac{\Delta_\ell}{4} \geq \mu_1 - \frac{\Delta_\ell}{2}\right)$$

$$= \Pr\left(\theta_1(\tau) \geq \hat{\mu}_1(\tau) - \frac{\Delta_\ell}{4}\right)\Pr\left(\hat{\mu}_1(\tau) - \frac{\Delta_\ell}{4} \geq \mu_1 - \frac{\Delta_\ell}{2}\right)$$

$$\overset{(c)}{\geq} \left(1 - \frac{1}{4}\exp(-n_1(\tau)\Delta_\ell^2/32)\right)\left(1 - \exp(-n_1(\tau)\Delta_\ell^2/8)\right)$$

$$= 1 - \frac{1}{4}\exp(-n_1(\tau)\Delta_\ell^2/32) - \exp(-n_1(\tau)\Delta_\ell^2/8) + \frac{1}{4}\exp(-5n_1(\tau)\Delta_\ell^2/32),$$
(8)

where the inequality $(c)$ is from Fact 1 and 2. Similarly, we can derive

$$\Pr\left(\theta_\ell(\tau) \leq \mu_\ell + \frac{\Delta_\ell}{2}\right) \geq 1 - \frac{1}{4}\exp(-n_\ell(\tau)\Delta_\ell^2/32) - \exp(-n_1(\tau)\Delta_\ell^2/8) + \frac{1}{4}\exp(-5n_\ell(\tau)\Delta_\ell^2/32).$$

Then we can derive $(b)$ using Fact 1 and we have ensured each arm is pulled at least 2 times during the warm-up round $t_0$.

For every $S \subseteq [t - t_0]$, we have an upper bound value $\delta_{\max}$ for $\Pr(X_\tau = 1 | \mathcal{F}_\tau)$: $\delta_{\max} = (N-1)(\frac{1}{2}\exp(-\Delta_{\min}^2/16) + 2\exp(-\Delta_{\min}^2/4) - \frac{1}{2}\exp(-5\Delta_{\min}^2/16) + \exp\left(-\frac{(t_0-1)\Delta_{\min}^2}{2N}\right) + \exp(-\Delta_{\min}^2))$. Let $\delta = \frac{1}{t-t_0}\sum_{\tau=t_0+1}^{t}\delta_{\max} = \delta_{\max}$ and $\gamma = \frac{(N-1)}{N}$, we can derive the following bound from Fact 3:

$$\Pr(n_1(t) < \frac{t-1}{N}) = \Pr(\sum_{\tau=0}^{t}X_\tau > t\frac{(N-1)}{N}) \leq \exp\left(-tD(\gamma\|\delta)\right).$$
(9)

$\square$

**Lemma 4.** *After the warm-up round $t_0$, for any sub-optimal arm $k \neq 1, \Delta_k = \mu_1 - \mu_k \geq 0$, the following inequality holds,*

$$\sum_{t=t_0+1}^{T}\Pr\left(k = k^{\text{emp}}(t), n_1(t) \geq \frac{(t-1)}{N}\right) \leq \frac{4N}{\Delta_k^2}$$

*Proof.* We bound the probability by :

$$= \sum_{t=t_0+1}^{T}\Pr\left(k = k^{\text{emp}}(t), n_1(t) \geq \frac{(t-1)}{N}\right)$$

$$\overset{(d)}{=} \sum_{t=t_0+1}^{T}\Pr\left(k = k^{\text{emp}}(t), n_1(t) \geq \frac{(t-1)}{N}, n_k(t) \geq \frac{(t-1)}{N}\right)$$

$$\leq \sum_{t=t_0+1}^{T}\Pr\left(\hat{\mu}_k(t) \geq \hat{\mu}_1(t), n_k(t) \geq \frac{(t-1)}{N}, n_1(t) \geq \frac{(t-1)}{N}\right)$$

$$\leq \sum_{t=t_0+1}^{T}\Pr\left(\left((\hat{\mu}_1(t) \leq \mu_1 - \frac{\Delta_k}{2})\bigcup(\hat{\mu}_k(t) \geq \mu_1 - \frac{\Delta_k}{2})\right), n_k(t) \geq \frac{(t-1)}{N}, n_1(t) \geq \frac{(t-1)}{N}\right)$$

$$= \sum_{t=t_0+1}^{T}\Pr\left(\left((\hat{\mu}_1(t) \leq \mu_1 - \frac{\Delta_k}{2})\bigcup(\hat{\mu}_k(t) \geq \mu_k + \frac{\Delta_k}{2})\right), n_k(t) \geq \frac{(t-1)}{N}, n_1(t) \geq \frac{(t-1)}{N}\right)$$

$$\overset{(e)}{\leq} \sum_{t=t_0+1}^{T}\Pr\left(\hat{\mu}_1(t) - \mu_1 \leq -\frac{\Delta_k}{2}, n_1(t) \geq \frac{(t-1)}{N}\right) + \sum_{t=t_0+1}^{T}\Pr\left(\hat{\mu}_k(t) - \mu_k \geq \frac{\Delta_k}{2}, n_k(t) \geq \frac{(t-1)}{N}\right)$$

$$\overset{(f)}{\leq} \sum_{t=t_0+1}^{T}2\exp\left(\frac{-(t-1)\Delta_k^2}{2N}\right) \overset{(g)}{\leq} \frac{4N}{\Delta_k^2},$$
(10)

Here, $(d)$ holds because of the truth that the empirical best arm is $k^{emp}(t)$ selected from the set $\mathcal{S}_t = \{k \in [N] : n_k(t) \geq \frac{(t-1)}{N}\}$. Inequality $(e)$ follows the union bound. We have $(f)$ from the truth that $\hat{\mu}_k(t) = \frac{\sum_{\tau=1}^{t} r_k(\tau)\mathbb{1}(k(\tau)=k)}{n_k(t)}, \forall k \in [N]$ and Fact 1. The last inequality $(g)$ uses the fact that $\frac{\Delta_k^2}{2N} > 0$ and the geometric series. $\hfill\square$

**Proof of Lemma 1.** Now, we prove Lemma 1 in the main paper.

*Proof.* During $t_0$ warm-up rounds, the maximum pulling times of a non-competitive arm $k^{nc}$ are bound in $t_0$. We then analyze the expected number of times pulling $k^{nc}$ after round $t_0$.

$$
\sum_{t=t_0+1}^{T} \Pr(k(t) = k^{nc})
$$

$$
= \sum_{t=t_0+1}^{T} \Pr(k(t) = k^{nc}, n_1(t) \geq \frac{(t-1)}{N}) + \sum_{t=t_0+1}^{T} \Pr(k(t) = k^{nc}, n_1(t) < \frac{(t-1)}{N})
$$

$$
\overset{(h)}{\leq} \sum_{t=t_0+1}^{T} \Pr(k(t) = k^{nc}, k^{nc} = k^{emp}(t), n_1(t) \geq \frac{(t-1)}{N})
$$

$$
+ \sum_{t=t_0+1}^{T} \Pr\left(k(t) = k^{nc}, k^{nc} \in \mathcal{S}_t \setminus k^{emp}(t), n_1(t) \geq \frac{(t-1)}{N}\right) + \sum_{t=t_0+1}^{T} \Pr(n_1(t) < \frac{(t-1)}{N})
$$

$$
\overset{(i)}{\leq} \sum_{t=t_0+1}^{T} \Pr\left(\hat{\mu}_1(t) \leq \hat{\varphi}_{k^{nc}}(t), k(t) = k^{nc}, n_1(t) \geq \frac{(t-1)}{N}\right) + \frac{4N}{\Delta_{k^{nc}}^2} + \sum_{t=t_0+1}^{T} \exp\left(-tD(\gamma\|\delta)\right)
$$

$$
\leq \sum_{t=t_0+1}^{T} \Pr\left(\left((\hat{\mu}_1(t) \leq \mu_1 + \frac{\tilde{\Delta}_{k^{nc},1}}{2}) \bigcup (\hat{\varphi}_{k^{nc}}(t) \geq \mu_1 + \frac{\tilde{\Delta}_{k^{nc},1}}{2})\right), k(t) = k^{nc}, n_1(t) \geq \frac{(t-1)}{N}\right)
$$

$$
+ \sum_{t=t_0+1}^{T} \exp\left(-tD(\gamma\|\delta)\right) + \frac{4N}{\Delta_{k^{nc}}^2}
$$

$$
\overset{(j)}{\leq} \sum_{t=t_0+1}^{T} \Pr\left(\hat{\mu}_1(t) \leq \mu_1 + \frac{\tilde{\Delta}_{k^{nc},1}}{2} n_1(t) \geq \frac{(t-1)}{N}\right)
$$

$$
+ \sum_{t=t_0+1}^{T} \Pr\left(\hat{\varphi}_{k^{nc}}(t) \geq \frac{\sum_{t=1}^{T} \mathbb{E}[r_{k^{nc}}^{max}(t)]}{T} - \frac{\tilde{\Delta}_{k^{nc},1}}{2}, k(t) = k^{nc}\right) + \sum_{t=t_0+1}^{T} \exp\left(-tD(\gamma\|\delta)\right) + \frac{4N}{\Delta_{k^{nc}}^2}
$$

$$
\overset{(k)}{\leq} \sum_{t=t_0+1}^{T} \exp\left(\frac{-(t-1)\tilde{\Delta}_{k^{nc},1}^2}{2N}\right) + \sum_{j=1}^{T} \Pr\left(\hat{\varphi}_{k^{nc}}(\tau_j) - \frac{\sum_{t=1}^{T} \mathbb{E}[r_{k^{nc}}^{max}(t)]}{T} \geq -\frac{\tilde{\Delta}_{k^{nc},1}}{2}\right)
$$

$$
+ \sum_{t=t_0+1}^{T} \exp\left(-tD(\gamma\|\delta)\right) + \frac{4N}{\Delta_{k^{nc}}^2}
$$

$$
\overset{(l)}{\leq} \sum_{t=t_0+1}^{T} \exp\left(\frac{-(t-1)\tilde{\Delta}_{k^{nc},1}^2}{2N}\right) + \sum_{j=1}^{T} \exp\left(-\frac{j\tilde{\Delta}_{k^{nc},1}^2}{2}\right) + \sum_{t=t_0+1}^{T} \exp\left(-tD(\gamma\|\delta)\right) + \frac{4N}{\Delta_{k^{nc}}^2}
$$

$$
\overset{(m)}{\leq} \frac{2N}{\tilde{\Delta}_{k^{nc},1}^2} + \frac{2}{\tilde{\Delta}_{k^{nc},1}^2} + \frac{1}{D(\gamma\|\delta)} + \frac{4N}{\Delta_{k^{nc}}^2} = \mathcal{O}(1),
$$

$$\tag{11}$$

Here, both $(h)$ and $(j)$ are derived using the union bound. We have $(i)$ from the Lemma 4 and Lemma 3. The inequality $(k)$ is obtained from Fact 1, wherein $j$ in $(k)$ explicitly denotes the round index when arm $k^{sub}$ is pulled. Inequality $(l)$ stems from Fact 1. We have $(m)$ because of the truth that $\frac{\tilde{\Delta}_{k^{nc},1}^2}{2N} \geq 0, \frac{\tilde{\Delta}_{k^{nc},1}^2}{2} \geq 0$, and $D(\gamma\|\delta) \geq 0$. We also use geometric series in $(m)$.

$\square$

We provide another Lemma to facilitate the proof of Lemma 2 in the main paper.

**Lemma 5.** *The following inequality holds,*

$$\Pr\left(E_2(t)\right) \leq 4(N-1)t\exp\left(-\frac{(t-1)\Delta_{min}^2}{2N}\right) + \frac{(N-1)}{D(\gamma\|\delta)},$$

*where $\Delta_{\min} = \min_k \Delta_k$.*

*Proof.*

$$\Pr\left(E_2(t)\right) \overset{(n)}{\leq} \sum_{\ell\in[N]\backslash 1} \Pr\left(\hat{\varphi}_1(t) < \hat{\mu}_\ell(t), n_1(t) \geq \frac{(t-1)}{N}, n_\ell(t) \geq \frac{(t-1)}{N}\right)$$

$$+ \sum_{\ell\in[N]\backslash 1} \Pr\left(\hat{\varphi}_1(t) < \hat{\mu}_\ell(t), n_1(t) < \frac{(t-1)}{N}, n_\ell(t) \geq \frac{(t-1)}{N}\right)$$

$$+ \sum_{\ell\in[N]\backslash 1} \Pr\left(\hat{\mu}_1(t) < \hat{\mu}_\ell(t), n_1(t) \geq \frac{(t-1)}{N}, n_\ell(t) \geq \frac{(t-1)}{N}\right)$$

$$\leq \sum_{\ell\in[N]\backslash 1} \Pr\left(\left((\hat{\varphi}_1(t) < \mu_1 - \frac{\Delta_{min}}{2})\bigcup(\hat{\mu}_\ell(t) > \mu_1 - \frac{\Delta_{min}}{2})\right),\right.$$

$$n_1(t) \geq \frac{(t-1)}{N}, n_\ell(t) \geq \frac{(t-1)}{N})$$

$$+ \sum_{\ell\in[N]\backslash 1} \Pr\left(\left((\hat{\mu}_1(t) < \mu_1 - \frac{\Delta_{min}}{2})\bigcup(\hat{\mu}_\ell(t) > \mu_1 - \frac{\Delta_{min}}{2})\right),\right.$$

$$n_1(t) \geq \frac{(t-1)}{N}, n_\ell(t) \geq \frac{(t-1)}{N}) + \sum_{\ell\in[N]\backslash 1} \Pr\left(n_1(t) < \frac{(t-1)}{N}\right)$$

$$\overset{(o)}{\leq} \sum_{\ell\in[N]\backslash 1} \Pr\left((\hat{\varphi}_1(t) < \frac{\sum_{t=1}^T \mathbb{E}[r_1^{\max}(t)]}{T} - \frac{\Delta_{min}}{2}, n_1(t) \geq \frac{(t-1)}{N}\right)$$

$$+ \sum_{\ell\in[N]\backslash 1} \Pr\left((\hat{\mu}_1(t) < \mu_1 - \frac{\Delta_{min}}{2}, n_1(t) \geq \frac{(t-1)}{N}\right)$$

$$+ 2\sum_{\ell\in[N]\backslash 1} \Pr\left(\hat{\mu}_\ell(t) > \mu_\ell + \frac{\Delta_{min}}{2}, n_\ell(t) \geq \frac{(t-1)}{N}\right) + (N-1)\exp\left(-tD(\gamma\|\delta)\right)$$

$$\overset{(p)}{\leq} 4(N-1)\exp\left(-\frac{(t-1)\Delta_{min}^2}{2N}\right) + (N-1)\exp\left(-tD(\gamma\|\delta)\right),$$

(12)

Inequality $(n)$, using union bound, arises from the observation that when arm 1 is absent from the empirical competitive set $\mathcal{E}^t$ at round $t$, it is either not selected as the empirical best arm $k^{emp}(t)$ or its $\hat{\varphi}_1(t)$ is less than the estimated mean of the empirical best arm $\hat{\mu}_{k^{emp}(t)}(t)$. The validity of inequality $(n)$ relies on the fact that $\frac{\sum_{t=1}^T \mathbb{E}[r_1^{\max}(t)]}{T} \geq \mu_1$ and Lemma 3. We establish the final inequality $(p)$ by leveraging Fact 1. $\square$

**Proof of Lemma 2.** Now, we present proof details of Lemma 2 in the main paper.

*Proof.* We split the analysis of $\sum_{t=1}^T \Pr(k(t) = k^{sub})$ into three parts: the pulls in the warm round; the pulls when the event $E_2(t)$ happens after the warm round; the pulls when the complementary of

$E_2(t)$ happens. We summarize it as follows:

$$\sum_{t=1}^{T} \Pr\left(k(t) = k^{sub}\right) = \sum_{t=1}^{t_0} \Pr\left(k(t) = k^{sub}\right) + \sum_{t=t_0+1}^{T} \Pr\left(k(t) = k^{sub}, E_2(t)\right)$$

$$+ \sum_{t=t_0+1}^{T} \Pr\left(k(t) = k^{sub}, E_2^c(t)\right) \tag{13}$$

$$\leq \sum_{t=1}^{T} \Pr\left(k(t) = k^{sub}\right) + \sum_{t=t_0+1}^{T} \Pr\left(E_2(t)\right)$$

When event $E_2(t)$ does not happen, the analysis of the upper bound of pulling the competitive but sub-optimal arm aligns to plain TS. We apply the result from Agrawal and Goyal (2012), which bounds the number of times a sub-optimal arm $k \neq 1$ is pulled within $\mathcal{O}(\log(T))$. In Lemma 5, when $E_2(t)$ happens, we derive the following bound:

$$\sum_{t=t_0+1}^{T} \Pr(E_2(t)) \leq \sum_{t=t_0+1}^{T} \left( 4(N-1)\exp\left(-\frac{(t-1)\Delta_{min}^2}{2N}\right) + (N-1)\exp\left(-tD(\gamma\|\delta)\right) \right)$$

$$\leq \frac{8N(N-1)}{\Delta_{min}^2} + \frac{1}{D(\gamma\|\delta)}$$

$$= \mathcal{O}(1). \tag{14}$$

The proof is completed. □

**Proof the Theorem 1.**

*Proof.* We revisit the definition of expected regret, given by:

$$\mathbb{E}[R(T)] = \mathbb{E}\left[\sum_{t=1}^{T}(\mu_1 - \mu_{k(t)})\right] = \mathbb{E}\left[\sum_{k=1}^{N} n_k(T)\Delta_k\right].$$

Considering $D$ competitive arms and $(N-D)$ non-competitive arms, the regret of E-TS in T rounds is bounded by:

$$\mathbb{E}[R(T)] = \sum_{k^{nc} \in [N-D]} \mathbb{E}[n_{k^{nc}}(T)]\Delta_{k^{nc}} + \sum_{k^{sub} \in [D]} \mathbb{E}[n_{k^{sub}}(T)]\Delta_{k^{sub}}$$

$$\overset{(1)}{\leq} \sum_{k^{nc} \in [N-D]} \Delta_{k^{nc}}\mathcal{O}(1) + \sum_{k^{sub} \in [D]} \Delta_{k^{sub}}\mathcal{O}(\log(T)) \tag{15}$$

$$\leq (N-D)\mathcal{O}(1) + D\mathcal{O}(\log(T)),$$

where the inequality $(1)$ is from Lemma 2 and Lemma 1. Thus the proof is finalized. □

## C  SUPPLEMENTARY EXPERIMENTS AND EXPERIMENTAL DETAILS

### C.1  DATASET AND MODEL STRUCTURE

Table 1 provides essential information about each dataset used in our study. We will introduce more details regarding the dataset characteristics and the corresponding model structures.

The **Credit** dataset consists of information regarding default payments, demographic characteristics, credit data, payment history, and credit card bill statements from clients in Taiwan. The dataset is partitioned evenly across six clients, each managing a bottom model with a Linear-BatchNorm-ReLU structure. The server hosts the top model, comprising of two Linear-ReLU-BatchNorm layers followed by a WeightNorm-Linear-Sigmoid layer.

The **Real-sim** dataset is from LIBSVM, which is a library for support vector machines (SVMs). 10 clients equally hold the data features and compute embeddings through a bottom model with 2 Linear-ReLU-BatchNorm layers. The server controls the top model with 3 Linear-ReLU layers.

The **FashionMNIST** dataset consists of $28 \times 28$ grayscale images of clothing items. The dataset is equitably distributed across 7 clients, with each holding a data portion of $28 \times 4$ dimensions. On the client side, it holds a Linear-BatchNorm-ReLU bottom model. On the server side, the top model comprises eight groups of Conv-BatchNorm-ReLU structures, two MaxPool layers, two Linear-Dropout-ReLU layers, and a final Linear output layer.

The **CIFAR-10** dataset contains 60,000 color images of size $32 \times 32$, representing vehicles and animals. We divide each image into $4 \times 32$ sub-images and distribute them among 8 clients. Each client's bottom model consists of 2 convolutional layers and 1 max-pooling layer. The server's top model is built with 6 convolutional layers and 3 linear layers.

The **Caltech-7** dataset, a subset of seven classes from the Caltech-101 object recognition collection, is distributed across six clients. Each client is assigned one unique feature view, encompassing the Gabor feature, Wavelet moments (WM), CENTRIST feature, Histogram of Oriented Gradients (HOG) feature, GIST feature, and Local Binary Patterns (LBP) feature, respectively. Every client maintains a bottom model utilizing a Linear-BatchNorm-ReLU structure. At the server level, the top model comprises eight Linear-ReLU layers, two Dropout layers, and a final Linear output layer.

The **IMDB** dataset comprises 50,000 highly polarized movie reviews, each categorized as either positive or negative. For distributed processing across 6 clients, each review is divided into several sentences, and an equal number of these sentences are allocated to each client. Each client utilizes a Bert model without fine-tuning—at the bottom level to obtain an embedding with 512 dimensions. These embeddings are then input to the server's top model, which consists of two Linear-ReLU layers followed by a final Linear output layer.

Table 1: VFL dataset and parameters descriptions.

| Task | Tabular | | CV | | Multi-view | NLP |
|---|---|---|---|---|---|---|
| Dataset name | Credit | Real-sim | FashionMNIST | CIFAR10 | Caltech-7 | IMDB |
| Number of samples | 30,000 | 72,309 | 70,000 | 60,000 | 1474 | 50,000 |
| Feature size | 23 | 20,958 | 784 | 1024 | 3766 | - |
| Number of classes | 2 | 2 | 10 | 10 | 7 | 2 |
| Number of clients | 7 | 10 | 7 | 8 | 6 | 6 |
| Batchsize $B^t$ | 32 | 512 | 128 | 32 | 16 | 64 |
| Warm-up rounds $t_0$ | 50 | 50 | 80 | 80 | 80 | 40 |

## C.2 EXPERIMENTAL RESULT IN ABLATION STUDY

Additional experiments have been conducted across a variety of datasets under diverse corruption constraints, as illustrated in Figure 5.

## C.3 DYNAMICS OF ARM SELECTION AND EMPIRICAL COMPETITIVE SET IN TS AND E-TS

We investigated the arm selection behavior of TS and E-TS during a targeted attack on FashionM-NIST, as shown in Figure 6. This study also tracked the variation in the size of E-TS's empirical competitive set, depicted in Figure 6. The parameters for this analysis were consistent with those in the FashionMNIST targeted attack scenario (Figure 1): $t_0 = 80$, $C = 2$, $\beta = 0.15$, $Q = 2000$ and the number of arms $N = \binom{7}{2} = 21$. We list all arms as follow:

[0: (client 1, client 2), 1: (client 1, client 3), 2: (client 1, client 4), 3: (client 1, client 5), 4:(client 1, client 6), 5: (client 1, client 7), 6: (client 2, client 3), 7: (client 2, client 4), 8: (client 2, client 5), 9: (client 2, client 6), 10: (client 2, client 7), 11: (client 3, client 4), 12: (client 3, client 5), 13: (client 3, client 6), 14: (client 3, client 7), 15: (client 4, client 5), 16: (client 4, client 6), 17: (client 4, client 7), 18: (client 5, client 6), 19: (client 5, client 7), 20: (client 6, client 7)].

Analysis of Figure 6(a) reveals that initially, E-TS selected a suboptimal arm. However, after 140 rounds, it consistently chose arm 5 (representing the pair of client 1 and client 7), indicating a stable

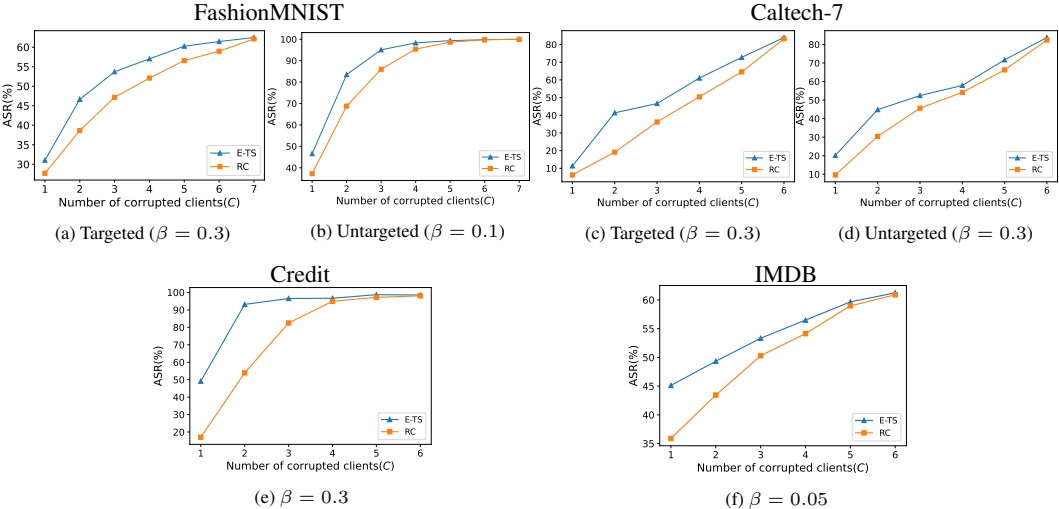

Figure 5: ASR using different number of corrupted clients.

selection. In contrast, TS continued to explore different arms during this period. Figure 6(b) shows that the empirical competitive set in E-TS reduced to a single arm within the first 40 rounds. Initially, the competitive arm selected by E-TS was not optimal. Nevertheless, E-TS effectively narrowed down its focus to this suboptimal arm, eventually dismissing it as non-competitive and identifying the best arm for selection.

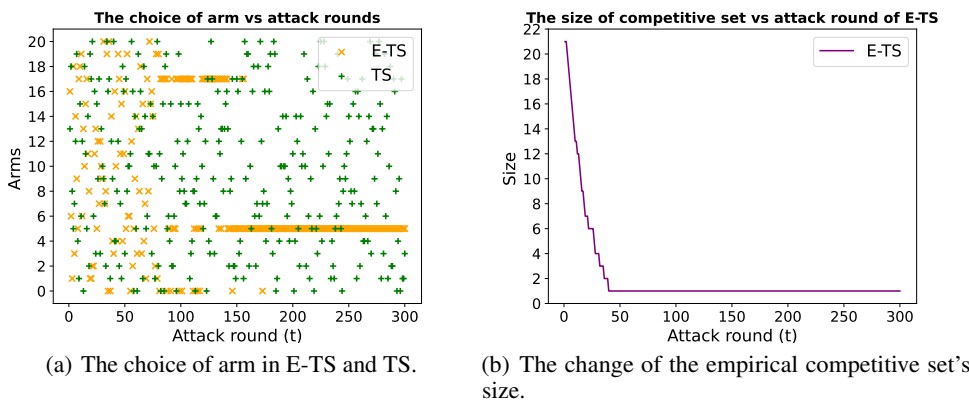

Figure 6: Dynamics of arm selection and competitive set in E-TS and TS.

## C.4 MINIMUM QUERY BUDGET AND CORRUPTION CHANNELS TO ACHIEVE 50% ASR

To explore how the necessary number of queries and corrupted channels vary across different models, datasets, and systems, we conducted experiments using Credit and Real-sim datasets. We specifically analyzed the average number of queries $q$ required to attain a 50% ASR under various levels of client corruption (corruption constraint $C$). For this analysis, we applied the proposed attack on both the Credit and Real-sim datasets in a 7-client setting. We varied $C$ from 1 to 7 and recorded the average queries $q$ needed for attacking over 50% of the samples successfully. In addition to assessing the impact of different datasets, we investigated the influence of model complexity by attacking two deeper Real-sim models contrasting it with the standard 3-layer server model. Specifically, the standard 3-layer model Real-sim(standard) has a Dropout layer after the first layer of the server model and achieves 96.1% test accuracy. One deeper server model Real-sim(deep) added an extra three layers to the Real-sim(standard) after the Dropout layer of the server model. Another model

Real-sim(dropout) structure is the same as Real-sim(deep) except that it added another Dropout layer before the penultimate layer of the server model. Both Real-sim(deep) and Real-sim(dropout) have 97% test accuracy. Furthermore, to analyze the system's effect on $q$ and $C$, we conducted experiments on Real-sim in a 10-client scenario, varying $C$ from 1 to 10 and recording $q$. Throughout these experiments, we maintained $\beta = 0.8$ and $t_0 = 2N$, where $N$ denotes the number of arms. The results are presented in Figure 7.

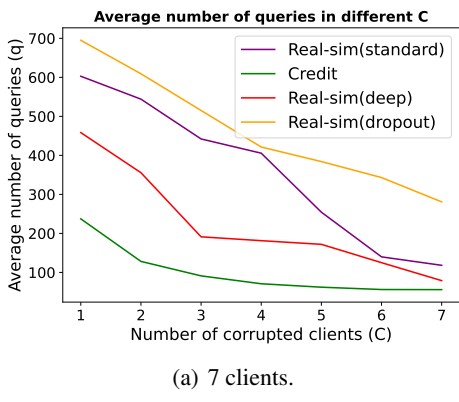
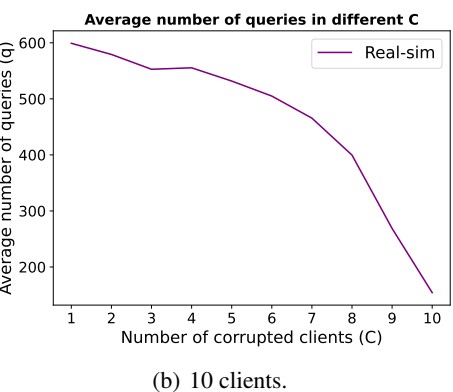

(a) 7 clients.

(b) 10 clients.

Figure 7: Average number of queries in different corruption constraint to achieve 50% ASR.

From Figure 7, we observe that the required average number of queries decreases with a looser (or higher) corruption constraint $C$. The comparison of Real-sim and Credit (Figure 7(a)) reveals that simpler datasets in the same task category (both being tabular datasets) necessitate fewer queries.

Contrary to our initial assumption, a deeper model does not necessarily require more queries. The results for Real-sim(standard), Real-sim(deep), and Real-sim(norm) from Figure 7(a) suggest that attacking a Real-sim(deep) requires fewer queries. A deeper model with an extra Dropout layer can make the model more robust and needs more quires to achieve 50% ASR. The reason for that is the deeper model will learn a different hidden feature of the sample, thus making the model have different robustness compared to the shallow one. Dropout can enhance robustness by preventing the model from becoming overly reliant on any single feature or input node, encouraging the model to learn more robust and redundant representations.

Comparing Figure 7 (a) and (b), we deduce that systems with more clients demand a greater number of queries to achieve the same ASR at a given $C$, due to each client possessing fewer features.

In conclusion, to attain a target ASR with the same $C$, simpler datasets within the same task require fewer queries. Systems with a higher number of clients necessitate more queries. However, the influence of the model's complexity does not simply depend on the scales of model parameters but is affected more by the Dropout layer.

### C.5 DISCUSSION ON THE LARGE EXPLORATION SPACES

We extend the experiments in Figure 4 to larger exploration spaces, i.e. set the corruption constraint $C = 7$ and $C = 8$, which results in $\binom{16}{7} = 11,440$, $\binom{16}{8} = 12,870$ arms, respectively. However, constrained by the computation power and limited time in the rebuttal period, we compare E-TS and plain TS in large exploration spaces through numerical simulation where ASR is substituted with a sample in Gaussian distribution. For the simulation, we created a list of means starting from 0 up to 0.99, in increments of 0.01, each with a variance of 0.1. This list was extended until it comprised $11,440 - 1$ and $12,870 - 1$ elements, to which we added the best arm, characterized by a mean of 1 and a variance of 0.1. This list represents the underlying mean and variance of the arms. Upon playing an arm, a reward is determined by randomly sampling a value, constrained to the range $[0, 1]$. With knowledge of the underlying mean, we plotted the cumulative regret over rounds, $R(t) = \sum_{\tau=1}^{t} (\mu_1 - \mu_{k(\tau)})$, where $\mu_1$ is the best arm's mean, and $k(\tau)$ is the arm selected in round $\tau$. These results are presented in Figure 8.

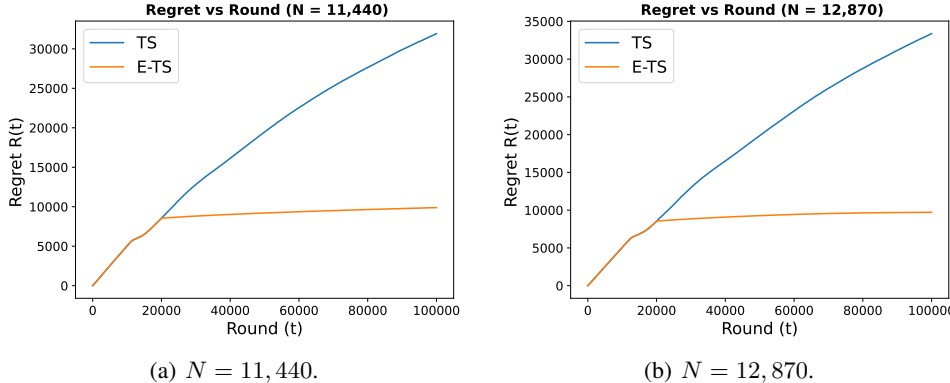

(a) $N = 11,440$.  (b) $N = 12,870$.

Figure 8: Regret in large exploration spaces.

The results from Figure 8 reveal that in large exploration spaces, TS struggles to locate the best arm within a limited number of rounds. In contrast, E-TS demonstrates more rapid convergence, further confirming the benefits of utilizing an empirical competitive set in large exploration spaces.

### C.6 THE STUDY OF OPTIMAL CHOICE ON THE WARM-UP ROUND $t_0$

To ascertain the ideal number of warm-up rounds $t_0$ for different arm settings, we conducted numerical experiments with $N = 100$ and $N = 500$. For $N = 100$, we experimented with $t_0 = 150$ (less than $2N$), $t_0 = 200, 300, 500$ (within $[2N, 5N]$), and $t_0 = 800$ (greater than $5N$). Similarly, for $N = 500$, the settings were $t_0 = 750$ (less than $2N$), $t_0 = 1000, 2000, 2500$ (within $[2N, 5N]$), and $t_0 = 4000$ (greater than $5N$).

In these experiments, ASR was replaced with Gaussian distribution samples. We initialized 100 arms with means from 0 to 0.99 (in 0.01 increments) and variances of 0.1. The reward for playing an arm was sampled from its Gaussian distribution. The cumulative regret $R(T)$ was computed as $R(T) = \sum_{t=1}^{T} (\mu_1 - \mu_{k(t)})$, where $\mu_1 = 0.99$ and $k(t)$ is the arm selected at round $t$, as illustrated in Figure 9.

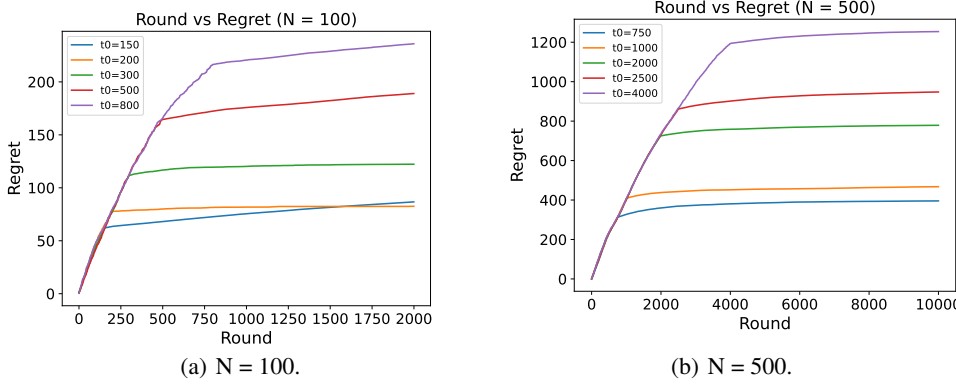

(a) N = 100.  (b) N = 500.

Figure 9: E-TS performance using different warm-up rounds.

Figure 9(a) shows that E-TS converges faster with a smaller $t_0$, but with $t_0 = 150$, it converges to a sub-optimal arm. Figure 9(b) indicates faster convergence with smaller $t_0$. Both figures suggest that $t_0 = 2N$ achieves the most stable and rapid convergence, while $t_0 > 5N$ results in the slowest convergence rate. Analyzing the pull frequencies of each arm during $t_0$, we find that with $t_0 < 2N$, most arms are pulled only once, and some are never explored. Conversely, with $t_0 \in [2N, 5N]$, most arms are pulled at least twice, yielding a more reliable estimation of their prior distributions.

Thus, we recommend setting $t_0$ to at least $N$, with the optimal range being $[2N, 5N]$ in practical scenarios. This range ensures that each arm is sampled at least twice using TS, enabling a more accurate initial assessment of each arm's prior distribution. Such preliminary knowledge is vital for E-TS to effectively form an empirical competitive set of arms. If $t_0$ is too small, there's an increased risk of E-TS prematurely converging on a suboptimal arm due to inadequate initial data, possibly overlooking the best arm in the empirical competitive set.

