# OpenReview forum: "Constructing Adversarial Examples for Vertical Federated Learning: Optimal Client Corruption through Multi-Armed Bandit"
_ICLR.cc/2024/Conference — ICLR 2024 poster_

### Official Review · Reviewer_qKnn · 2023-10-31

**Soundness:** 3 good
**Presentation:** 3 good
**Contribution:** 2 fair
**Rating:** 5
**Confidence:** 4

**Summary:**

To me it seems the main contribution of this paper is the formulation of the adversarial attacks in the VFL setting as a MAB problem and the development of the E-TS algorithm, whose usage in practice seems unrealistic to me, considering the combinatorial structure of the problem and availability of combinatorial MAB algorithms.

**Strengths:**

The paper is nicely written and executed.
Their contributions are clear and the experimental results are fairly diverse.
The formulations of the problem in (3) and (4) as a MAB problem is interesting, and to my knowledge and as per authors' claim, novel as well.
The E-TS algorithm is an interesting approach to circumvent the combinatorial structure of the problem and the authors provide regret bounds and experimental evidence as to the expected performance improvement gained from E-TS. However, I am not fully convinced to its practical utility.

**Weaknesses:**

The E-TS algorithm is supposed to outperform the TS algorithm via the machinery of "competitive arm set" constructed at each round.

A critical point that leaves me in doubt about E-TS is the number of required warm-up rounds t_0 being too large to the point it renders E-TS useless, even with fairly small scale problems. This is suggested both by intuition and Lemma 2, i.e., it grows proportional to the inverse square of the minimum sub-optimality gap and to the number of total arms, which grows exponentially in number of individuals arms to be selected, therefore rendering E-TS' practical utility very limited in my opinion. For instance, I would like to see what happens when the number of clients to be selected is increased from 2 or 3 to, say, 7-8 in Figure 4. There is a vast literature on "combinatorial" MABs which could offer better solutions to this problem with lesser assumptions, e.g., by imposing and exploiting some sort of structure between the arms.

**Questions:**

no questions for now

---

> ### Author Response · Authors · 2023-11-18
> **Response to reviewer qKnn (Part 1/2)**
>
> Thank you to the reviewers for your thoughtful and detailed feedback on our paper. We value the time and effort you invested in reviewing our work, and we are grateful for your insightful observations. In this rebuttal, we will comprehensively address the key points you have highlighted. We are keen to provide clarifications and look forward to enhancing our paper with the guidance from your suggestions.
>
> ## Weakness 1:
>
> **Q:** The E-TS algorithm is supposed to outperform the TS algorithm via the machinery of "competitive arm set" constructed at each round.
> A critical point that leaves me in doubt about E-TS is the number of required warm-up rounds $t_0$ being too large to the point it renders E-TS useless, even with fairly small scale problems. This is suggested both by intuition and Lemma 2, i.e., it grows proportional to the inverse square of the minimum sub-optimality gap and to the number of total arms, which grows exponentially in number of individuals arms to be selected, therefore rendering E-TS' practical utility very limited in my opinion.
>
> **R:** We appreciate the reviewer’s perspective on the practicality of the E-TS algorithm, particularly concerning the number of warm-up rounds, $t_0$. We have added the discussion of the choice on warm-up rounds $t_0$ in **Appendix C.4**.
>
> To address this, first, it's important to clarify the relationship between $t_0$ and the total number of arms $N$. According to our formulation, $t_0$ needs to satisfy the condition $\frac{t_0}{\log(t_0)} \geq \frac{8N}{\Delta_{\min}^2}$. This indicates a superlinear relationship with $N$, but not necessarily an exponential one.
>
> In practical scenarios, setting $t_0$ to be at least larger than the number of arms $N$ is usually sufficient. Our experiments in **Figure 7 of Appendix C.4** indicate that a range between $2N$ and $5N$ is optimal. This range ensures that each arm is sampled at least twice using TS, providing the adversary with a more accurate initial understanding of each arm's prior distribution. This preliminary knowledge is crucial for the E-TS to effectively identify an empirical competitive set of arms. If $t_0$ is too small, there's a risk of E-TS prematurely converging on a suboptimal arm due to insufficient initial data, potentially omitting the optimal arm from the empirical competitive set.
>
> The optimal range of $t_0$ can be further validated in Figure 3 (b) in Section 7.2, where the number of arms $N =21$ and the optimal range of $t_0$ should be $[42, 105]$. , we can observe that if we set $t_0 = 20$, E-TS converges to a suboptimal arm, and when $t_0 = 80$ and $t_0 = 200$, E-TS can converge to the optimal arm. Specifically, E-TS with $t_0 = 80$ has faster convergence than the one with $t_0 = 200$ because, within 80 rounds, E-TS has enough information of each arm to construct the empirical competitive set, thus discarding the non-competitive arm.
>
> In conclusion, we recommend the choice of $t_0$ in the range of $[2N, 5N]$ , aiding in the accurate estimation of arm distributions and enhancing the overall effectiveness of E-TS (see Appendix C.4).

---

> ### Author Response · Authors · 2023-11-18
> **Response to reviewer qKnn (Part 2/2)**
>
> ## Weakness 2:
> **Q:** For instance, I would like to see what happens when the number of clients to be selected is increased from 2 or 3 to, say, 7-8 in Figure 4.
>
> **R:**  We understand the interest in exploring scenarios with a higher number of selected clients. However, increasing the number of clients leads to a significant expansion in the exploration space. Specifically, the exploration spaces for selecting 7 out of 16, 8 out of 16, 7 out of 28, and 8 out of 28 clients are $\binom{16}{7} = 11,440$, $\binom{16}{8} = 12,870$, $\binom{28}{7} = 1,184,040$, and $\binom{28}{8} = 3,108,105$, respectively.
>
> We extend the experiments in  Figure 4 to larger exploration spaces, i.e. set the corruption constraint $C = 7$ and $C = 8$, which results in $\binom{16}{7} = 11,440,\binom{16}{8} = 12,870$ arms, respectively. However, constrained by our available computation resources and the limited time of the rebuttal period, we compare E-TS and plain TS in large exploration spaces through numerical simulation where ASR is substituted with random reward following Gaussian distribution. Specifically, we generate 11,440 and 12,870 arms, which mean values are in the range $(0,1)$ (increments in 0.01) and variances is 0.1, respectively. Among arms, only arm 1 is the best arm with a mean of 1. Upon playing an arm, a reward is determined by randomly sampling a value from its Gaussian distribution and constrained to the range $[0,1]$. With knowledge of the underlying mean, we plotted the cumulative regret over rounds, $R(t) = \sum_{\tau=1}^t (\mu_1-\mu_{k(\tau)})$, where $\mu_1$ is the best arm's mean, and $k(\tau)$ is the arm selected in round $\tau$. These results are presented in **Figure 11 of Appendix F.**
>
> The results from Figure 11 reveal that in large exploration spaces, TS struggles to locate the best arm within a limited number of rounds. In contrast, E-TS demonstrates more **rapid convergence**, further confirming the benefits of utilizing an empirical competitive set in large exploration spaces.
>
> ## Weakness 3:
> **Q:** There is a vast literature on "combinatorial" MABs which could offer better solutions to this problem with lesser assumptions, e.g., by imposing and exploiting some sort of structure between the arms.
>
>
> **R:** We initially considered CMAB for the corruption pattern selection issue but ultimately chose MAB due to several reasons:
>
>
>
> 1.  **Clear Definition of Rewards:** In a CMAB framework, defining the reward for each base arm is challenging, especially under varying corruption constraints. For instance, using the average ASR (Attack Success Rate) of including this arm to represent a base arm's reward can result in inconsistencies across different constraints. MAB, on the other hand, allows for a straightforward definition of each arm's reward as the ASR when that particular arm (corruption pattern) is used.
>
>
>
> 2.  **Simplified Assumptions:** CMAB typically assumes the rewards of base arms to be independent and identically distributed (i.i.d.), which does not hold in our context due to the influence of corruption constraints. Moreover, common CMAB models [1] [2] assume that the super arm's reward is a simple aggregate (average or sum) of the base arms' rewards, which does not align with our attack model. Although some CMAB variants [3] [4] consider nonlinear relationships between super and base arms, they still rely on observable rewards for base arms or presume i.i.d. distributions, which are not applicable in our case.
>
>
>
> 3.  **Comparative Performance and Regret Bound:** The regret bound in most CMAB works is $\mathcal{O}(\log(T))$ [1] [2] [3] [4], similar to MAB solutions. However, our MAB approach improves the regret of non-competitive arms from $\mathcal{O}(\log(T))$ to $\mathcal{O}(\log(1))$, showing a significant advantage. Additionally, our initial trials with a standard CMAB algorithm [1] yielded poorer results compared to a plain Thompson Sampling (TS) approach in MAB.
>
>
>
> Therefore, considering these aspects, MAB was chosen for its clearer reward structure, less restrictive assumptions, and superior performance in our specific scenario.
>
>
>
> [1] Wang et al. (2018) Thompson Sampling for Combinatorial Semi-Bandits
>
>
> [2] Combe et al. (2015) Combinatorial Bandits Revisited
>
>
> [3] Chen et al. (2016) Combinatorial Multi-Armed Bandit with General Reward Functions
>
>
> [4] Perrault et al. (2020) Statistical Efficiency of Thompson Sampling for Combinatorial Semi-Bandits

---

> ### Author Response · Authors · 2023-11-20
> **Can you please response to our rebuttal for Paper5036**
>
> Dear Reviewer qKnn,
>
> We are approaching the end of the discussion phase, and we have unfortunately received no feedback from you on our rebuttal.
>
> Please can we kindly ask you to take a look at our responses, and let us know whether we have clarified your questions and addressed your concerns?
>
> Thank you very much again for the time you spent reviewing.
>
> Paper5036 Authors

---

### Official Review · Reviewer_XBEx · 2023-11-01

**Soundness:** 3 good
**Presentation:** 2 fair
**Contribution:** 2 fair
**Rating:** 8
**Confidence:** 4

**Summary:**

The paper considers adversarial attacks on Vertical Federated Learning via selection of a set of clients to corrupt, and choice of perturbations of their representations in VFL. The perturbations are subject to a bound, as also the cardinality of the selected client set.  The client subset selection problem (choosing C out of M clients to corrupt) is recast as a MAB problem, and a modified Thompson sampling approach is proposed to deal with the computational complexity (#arms Some theoretical results are presented, bounding the regret under the modified sampling method. ). Performance results are presented for several datasets.

**Strengths:**

VFL is an emerging area, and there has not been much work studying adversarial attacks in this setting.

**Weaknesses:**

The VFL setting as explained at the top of Section 2 is puzzling. This implies that a prediction query arrives simultaneously at all M clients.  But even in the given setting, why does the server need to broadcast the full probability vector, rather than the class with the maximal value? Is there a notion of a primary query client? Or is it more realistic that the query comes to the central server which then asks clients for their representations?

What was the loss function L (in eqn (3), etc.) adopted in the experimental settings?

Why is the constant \alpha^* needed in (4)? The later discussion in Sec 5.2 that the mean reward corresponding to the best arm (client subset) can be taken to be \alpha^* is also confusing. The adversary does not know \alpha^*, so how can it proceed to solve (4)?

Algorithm 1:
- Please identify the inputs: t_o, {B_t, t=1,..,T} and the corresponding embeddings …..
- After line 11, shouldn’t there be a call to Alg 2 to compute the corrupted embeddings?
- Line 6 should be in plural
- Line 13: are the rewards assumed to be positive?
- The role of \phi is not clear. Let n(t) be the number of times a given arm is played, and let r(j) denote the corresponding rewards. Then \phi = \frac{1}{n(t)} \sum_{j=1}^{n(t)} \max(r_i, i=1,…j). Why is this an useful quantity to compute
-  What is the reward? Is it merely the ASR over the batch? With reference to Remark 1 and the discussion of [Gupta et al., 2021], is the reward in this setting simply 0 or 1 (so max=1)?

Experiments
Given that the value of N is small in all the settings (M takes values of 6, 7, 8 and 10, and C = 2,3,4), it would have been insightful to show how the best arm selection changes over t \in [1, T]. It would also have been insightful to show how the size of the restricted set decrease with t.

The choice of datasets is poorly motivated for the VFL problem. The credit score and Caltech datasets are reasonable, but segmenting a CIFAR or MNIST image into segments, or an IMDB review at different clients is not reasonable.

A value of Q=1000 is outrageously large. Servers with any reasonable security settings would turn off queries if they are repeated more than 3 or 5 times. A large Q is simply a mechanism for an adversary to learn (more samples). Q = 1000 is not a reasonable number.
What were the default values of \beta, B^t, T and C?

The “Ablation study” is not particularly helpful. That the adversary’s success rate would increase with \beta, T, or C is obvious. This does not add any insights. Further discussion on choosing t_o would be useful.

In Figure 1 what do the curves ‘Client 4 5’ etc. mean? Why are different client pairs shown in different plots? What is a test epoch and what does 25 attack rounds/test epoch mean? Does this mean that B^t is fixed at 25? And that T  was 30 (judging from the figures)?
There is little difference between TS and E-TS after about 5 epochs. What is the corresponding complexity comparison? The log(T) to O(1) improvement does not seem to show up here.

Figure 2: Fashion MNIST has 7 clients; so even with all 7 clients corrupted, manifold defense does pretty well. This is a surprising and unexpected result.

Figures 2, 3 (and parts of 1): Figures showing performance results on Credit Score or the Caltech dataset would have been more meaningful. More specifically, recognition of MNSIT / Fashion MNIST / CIFAR does not require all pieces of the full image; neither does sentiment analysis of a review; further, as noted earlier, such segmentation is unrealistic in any case.

Figure 4: What is the setting with 16 and 28 clients? How were the images segmented? Is it meaningful to have a 28-pixel image segment at each of 28 clients?

Theoretical results
Lemma 1, Lemma 2, and Theorem 1 appear to have an implicit large N assumption.  The lower bound on the probability is negative for N < 55, and exceeds 0.5 at about N=150.

How does the proposed E-TS algorithm work compared to sequential arm elimination proposed in
S. Shahrampour, M. Noshad and V. Tarokh. On Sequential Elimination Algorithms for Best-Arm Identification in Multi-Armed Bandits. IEEE Transactions on Signal Processing, vol. 65, no. 16, pp. 4281-4292, 15 Aug.15, 2017

Is a factor 2 missing in the equation for Fact 1?

In the proof of Lemma 4, shouldn’t the = sign be <= on line 5. The transition from line 6 to line 7 (“g”) is not clear.

Proof of Lemma 5: The fact that … towards the end of the proof is not obvious; it holds asymptotically; why does it hold for all T?

Proof of Theorem 1: Please add proof details. The terse statements are insufficient.

Puzzling statement in Section 3: “β ∈ [0, 1] is the perturbation budget of some simple magnitude-based anomaly detector.” Whose anomaly detector, and why is it assumed to be simple?



Before (2): “taking over” should be “taken over”.

P6, line 7: “obtain” should be “obtains”.

The sentence about the large exploration space (below Fig 1 and Fig 2) is confusing. How are these numbers obtained?

**Questions:**

Please see comments under Weaknesses

--- Added after rebuttal/discussion phase -

The authors have addressed all my major concerns; they have added substantially to the Supplementary Material, and made changes as needed, including strengthening of one of the Lemmas. Based on this I have changed my score to Accept.

---

> ### Author Response · Authors · 2023-11-18
> **Response to reviewer XBEx (Part 1/8)**
>
> We are deeply grateful to the reviewer for the careful and patient evaluation of our paper. Your detailed and insightful feedback not only demonstrates your dedication to the field but also significantly contributes to the enhancement of our work. In this rebuttal, we aim to thoughtfully address each of the points you have raised, reflecting on your valuable insights. We believe that your meticulous review has offered us a unique opportunity to refine and strengthen our research.
>
> ## Question 1:
> **Q:** The VFL setting as explained at the top of Section 2 is puzzling. This implies that a prediction query arrives simultaneously at all M clients. But even in the given setting, why does the server need to broadcast the full probability vector, rather than the class with the maximal value? Is there a notion of a primary query client? Or is it more realistic that the query comes to the central server which then asks clients for their representations?
>
>
> **R:** In our VFL framework, the server's role is akin to a **remote model/API with a limited query budget**, as outlined in prior works [1] [2] [3]. The query process is initiated by **arbitrarily one of the corrupted clients**, which then collaborates with other clients to submit their embeddings for prediction. There is **no designated 'primary query client'** in this model.
>
> Regarding the transmission of the full probability vector, our approach aligns with previous studies [1] [2] [3], where an API/remote model query returns various forms of outputs, including probability vectors, prediction probabilities, or hard labels. In our threat model, the server broadcasts the probability vector to all VFL clients. The probability vectors are used to compute the loss in the Inner problem AEG, as shown in equation 3 of section 5. Specifically, in both targeted and untargeted attacks, the adversary calculates the cross-entropy between the probability vector and the target label. It's noteworthy that the essence of our problem formulation and the E-TS solution remains **unaffected** even if the server provides only hard labels or prediction probabilities. In such cases, alternate forms of loss functions can be employed to address the inner problem.
>
> [1] Ilyas et al. (2018) Black-box Adversarial Attacks with Limited Queries and Information.
>
> [2]Yu et al. (2020) CloudLeak: Large-Scale Deep Learning Models
> Stealing Through Adversarial Examples
>
> [3] Chen et al. (2020) HopSkipJumpAttack: A Query-Efficient
> Decision-Based Attack
>
>
>
>
> ## Question 2:
> **Q:**
>   What was the loss function L (in eqn (3), etc.) adopted in the experimental settings?
>
> **R:** We consider the loss function $L(\boldsymbol{\eta}\^t\_i; \mathcal{C}^t) = l(\boldsymbol{f}\_0(\tilde{\boldsymbol{h}}^t\_{i,a};\boldsymbol{h}\_{i,b}^t), y\_v)$ for the targeted attack with target label $y\_v$, and $L(\boldsymbol{\eta}^t\_{i}; \mathcal{C}^t) = -l(\boldsymbol{f}\_0(\tilde{\boldsymbol{h}}^t\_{i,a};\boldsymbol{h}\_{i,b}^t), y\_u)$ for the untargeted attack with label $y_u$, where $l(\cdot)$ denotes the loss metric, and $\boldsymbol{f}\_0(\tilde{\boldsymbol{h}}^t\_{i,a};\boldsymbol{h}\_{i,b}^t)$ returns the probability vectors of the test sample $i$. In our experiments, we use cross entropy as the loss metric $l(\cdot)$.
>
>
>
>
>
>
> ## Question 3:
>
> **Q:** Why is the constant $\alpha^*$ needed in (4)? The later discussion in Sec 5.2 that the mean reward corresponding to the best arm (client subset) can be taken to be $\alpha^*$ is also confusing. The adversary does not know $\alpha^*$, so how can it proceed to solve (4)?
>
> **R:** The inclusion of $\alpha^*$ in function (4) is primarily to facilitate a coherent **transformation** to the standard MAB problem as outlined in function (1). In this context, $\alpha^*$ can be any positive constant, as it does not directly influence the optimization goal of the CPS problem, which is to maximize the expected Attack Success Rate (ASR) by selecting an appropriate corruption pattern $\mathcal{C}^t$, i.e., $\max_{\mathcal{C}^t}\mathbb{E}_t\left[A^*(\mathcal{C}^t;{B^t})\right]$. In the standard MAB formulation, the goal is to minimize the regret, which is defined as:
>
> $\mathbb{E} [R(T)] = \mathbb{E} \left[\sum_{t=1}^T (\mu^* - \mu_{k(t)})\right] = \mathbb{E} \left[\sum_{k=1}^N n_k(T)\Delta_k\right].$
>
> Therefore, we can set $\alpha^* = \mu^*$ to establish the equivalence of the problems and accomplish the transformation.
>
> It's important to note that while the adversary may not have prior knowledge of $\alpha^*$, this does not impede the process of solving (4). From regret definition, minimizing the gap between $\mu^*$ and $\mu_{k(t)}$ can be transformed to minimize the pulling times of any suboptimal arm, i.e., $n_k(T), k\neq k^*$. In our proposed attack, we solve (4) by identifying and playing the best arm and discarding any sub-optimal arm though the E-TS algorithm.

---

> ### Author Response · Authors · 2023-11-18
> **Response to reviewer XBEx (Part 2/8)**
>
> ## Question 4:
>
> **Q:** Algorithm 1:
>
> Please identify the inputs: $t_0, {B_t, t=1,..,T}$ and the corresponding embeddings …..
> After line 11, shouldn’t there be a call to Alg 2 to compute the corrupted embeddings?
> Line 6 should be in plural
> Line 13: are the rewards assumed to be positive?
> The role of $\phi$ is not clear. Let n(t) be the number of times a given arm is played, and let r(j) denote the corresponding rewards. Then $\phi = \frac{1}{n(t)} \sum_{j=1}^{n(t)} \max(r_i, i=1,…j)$. Why is this an useful quantity to compute
> What is the reward? Is it merely the ASR over the batch? With reference to Remark 1 and the discussion of [Gupta et al., 2021], is the reward in this setting simply 0 or 1 (so max=1)?
>
> **R:** Thank you for your detailed questions and suggestions. We provide the following clarifications and modifications:
>
> 1. Inputs to Algorithm 1: The variable $t_0$ denotes the number of warm-up rounds. $B^t$ represents the batch size at time $t$, with $[B^t]$ indicating the set of indices for the batch data. The corrupted embedding $\boldsymbol{h}_{i,a}^t$ is an aggregation of embeddings from multiple corrupted clients, where $a_1,\ldots, a_C$ are the indices of these clients in the corruption pattern $\mathcal{C}^t$.
>
> 2. Call to Algorithm 2: A call to Algorithm 2 for computing the corrupted embeddings is indeed necessary after line 11. This step has been incorporated into line 12 in the revised draft.
> 3. Pluralization and Rewards: The correction in line 6 to pluralize the term has been finished. Rewards are assumed to be positive, lying in the range of [0,1], as they represent the ASR of batch data.
> 4. Function of $\hat{\phi}\_k$: $\hat{\phi}\_k$ is defined as **the average of the maximum rewards observed for arm $k$**. It is utilized to **identify competitive arms** by comparing it with the empirical mean $\hat{\mu}_{k^{emp}(t)}$ of the empirical best arm $k^{emp}(t)$. Arms with $\hat{\phi}\_k > \hat{\mu}\_{k^{emp}(t)}$ are selected as competitive, aiding in reducing exploration space.
> 5. Nature of Reward: The reward in our model is the ASR of batch data for a given corruption pattern. It ranges from 0 to 1, with 1 being the maximum possible reward.
>
>
> ## Question 5:
> **Q:** Experiments Given that the value of N is small in all the settings (M takes values of 6, 7, 8 and 10, and C = 2,3,4), it would have been insightful to show how the best arm selection changes over $t \in [1, T]$. It would also have been insightful to show how the size of the restricted set decrease with t.
>
> **R:** We report the dynamics of arm selection and the size of the restricted set (empirical competitive set) **in Figure 8 of Appendix C**.5. The parameters for this analysis were consistent with those in the FashionMNIST targeted attack scenario (Figure 1): $t_0 = 80$, $C = 2$, $\beta = 0.15$, $Q = 2000$ and the number of arms $N = \binom{7}{2} = 21$.
>
> Analysis of Figure 8 (a) of Appendix C.5 reveals that initially, E-TS selected a suboptimal arm. However, after 140 rounds, it consistently chose arm 5 (representing the pair of client 1 and client 7), indicating a stable selection. In contrast, TS continued to explore different arms during this period. Figure 8 (b) of Appendix C.5 shows that the restricted set (empirical competitive set) in E-TS reduced to a single arm after around 40 rounds. Initially, the competitive arm selected by E-TS was not optimal. Nevertheless, E-TS effectively narrowed down its focus to this suboptimal arm, eventually dismissing it as non-competitive and identifying the best arm for selection.
>
> ## Question 6:
>
> **Q:** The choice of datasets is poorly motivated for the VFL problem. The credit score and Caltech datasets are reasonable, but segmenting a CIFAR or MNIST image into segments, or an IMDB review at different clients is not reasonable.
>
> **R:** We notice the unreasonable dataset. Since the scarcity of the VFL dataset, many previous works not only test VFL performance on tabular and multiview datasets but also on CV and NLP datasets by splitting the original data sample (see, e.g., [4] [5] [6]). We follow previous works to test the performance of our attack in distinct tasks. We hope to further investigate and apply more suitable dataset on CV and NLP tasks.
>
> [4] Chen et al. (2020) Vafl: a method of vertical asynchronous federated learning.
>
> [5] D Romanini et al. (2021) Pyvertical: A vertical federated learning framework for multi-headed splitnn.
>
> [6] Li et al. (2023) FedVS: Straggler-Resilient and Privacy-Preserving Vertical Federated Learning for Split Models.

---

> ### Author Response · Authors · 2023-11-18
> **Response to reviewer XBEx (Part 3/8)**
>
> ## Question 7:
>
> **Q:** A value of Q=1000 is outrageously large. Servers with any reasonable security settings would turn off queries if they are repeated more than 3 or 5 times. A large Q is simply a mechanism for an adversary to learn (more samples). Q = 1000 is not a reasonable number. What were the default values of $\beta, B^t, T$ and $C$?
>
> **R:** In our model, we conceptualize the server as an API, similar to how it's approached in previous works [1] [2] [3]. For instance, the Google Cloud Vision API allows a query budget of up to 1800 for a single sample. Our primary focus in this research is on the efficiency of the E-TS algorithm in **selecting the optimal corruption pattern**. The inner Adversarial Example Generation (AEG) problem aligns more closely with traditional adversarial example generation techniques.
>
> To address the concern about the large query budget, first, we conduct a targeted attack with the target label set to 6, under the setting that a query budget of 500, perturbation budget of 0.3, and a corruption constraint of 2, the E-TS best arm results indicate an ASR of 51\%, which indicates even in small query budget, we still can successfully attack half of the samples. Besides, adversaries could duplicate multiple noisy versions of the samples before inference, effectively increasing their query budget more stealthily. Furthermore, recent advancements in query-efficient methods [3] [7] and transfer-based approaches [8] offer alternatives to the AEG problem that do not compromise the efficacy of the E-TS solution.
>
> In sum, we can leverage methods, like **selecting suitable target labels and perturbation budgets, duplicating samples in advance, and using efficient strategies** to mitigate the concern of a limited query budget. We also conduct an analysis of necessary minimum number of queries in **Appendix D**.
> Regarding the default parameter settings used in our ablation study, they were as follows: $\beta = 0.3, B^t = 128$, and $C = 2$. We varied test parameters while maintaining the stability of other settings. The comprehensive parameter settings are now detailed in the revised draft and **Table 1 of Appendix C.1**.
>
> [1] Ilyas et al. (2018) Black-box Adversarial Attacks with Limited Queries and Information.
>
> [2]Yu et al. (2020) CloudLeak: Large-Scale Deep Learning Models
> Stealing Through Adversarial Examples
>
> [3] Chen et al. (2020) HopSkipJumpAttack: A Query-Efficient
> Decision-Based Attack
>
> [7] Andriushchenko et al. (2020) Square Attack: a query-efficient black-box
> adversarial attack via random search
>
> [8] Chen et al. (2023) Rethinking Model Ensemble in Transfer-based Adversarial Attacks
>
> ## Question 8:
>
> **Q:** The “Ablation study” is not particularly helpful. That the adversary’s success rate would increase with $\beta, T$, or $C$ is obvious. This does not add any insights. Further discussion on choosing $t_0$ would be useful.
>
> **R:** Thanks for the suggestions. We have now added related discussions in **Appendix C.4**. In practical scenarios, setting $t_0$ to be at least larger than the number of arms $N$ is usually sufficient. Our experiments indicate that a range between
> $2N $ and $5N$ is optimal. This range ensures that each arm is sampled at least twice using TS, providing the adversary with a more accurate initial understanding of each arm's prior distribution. This preliminary knowledge is crucial for the E-TS to effectively identify an empirical competitive set of arms. If $t_0$ is too small, there's a risk of E-TS prematurely converging on a suboptimal arm due to insufficient initial data, potentially omitting the optimal arm from the empirical competitive set.
>
> The optimal range of $t_0$ can be further validated in Figure 3 (b) in Section 7.2, where the number of arms $N =21$ and the optimal range of $t_0$ should be $[42, 105]$. , we can observe that if we set $t_0 = 20$, E-TS converges to a suboptimal arm, and when $t_0 = 80$ and $t_0 = 200$, E-TS can converge to the optimal arm. Specifically, E-TS with $t_0 = 80$ has faster convergence than the one with $t_0 = 200$ because, within 80 rounds, E-TS has enough information of each arm to construct the empirical competitive set, thus discarding the non-competitive arm.
>
>
> Therefore, the optimal selection of $t_0$ should be in range $[2N, 5N]$, aiding in the accurate estimation of arm distributions and enhancing the overall effectiveness of E-TS.

---

> ### Author Response · Authors · 2023-11-18
> **Response to reviewer XBEx (Part 4/8)**
>
> ## Question 9:
>
>
> **Q:** In Figure 1 what do the curves ‘Client 4 5’ etc. mean? Why are different client pairs shown in different plots? What is a test epoch and what does 25 attack rounds/test epoch mean? Does this mean that $B^t$ is fixed at 25? And that $T$ was 30 (judging from the figures)? There is little difference between TS and E-TS after about 5 epochs. What is the corresponding complexity comparison? The $log(T)$ to $O(1)$ improvement does not seem to show up here.
>
> **R:** We apologize for the confusion. The curve like 'Client 4,5' represents the attack performance of a fixed corruption pattern that corrupts client 4 and client 5 consistently in the inference. Similarly, in each sub-figure of Figure 1, curves like 'Client a,b,c' represent the performance of baseline method ''Fixed corruption pattern" mentioned in the Setup part of section 7. Specifically, we choose the best corruption pattern and randomly select a suboptimal corruption pattern for comparison, where the best corruption pattern differs in different attacks and datasets.
>
> The '25 attack rounds/test epoch' means that we report the ASR of 25 attack rounds and regard these test rounds as an epoch for plotting a smooth figure. For the concern of the relations of $B^t$, attack round, epoch, and $T$, each attack round contains $B^t$ samples, each epoch is composed of $n_a$ attack rounds, e.g., 25 attack rounds in FashionMNIST, such that contains $n_aB^t$ test samples if $B^t$ is same for all $t$, and $T$ actually equals to $30\times n_a$. Note that the adversary can adjust the corruption pattern after each attack round.
>
> For the concern of complexity comparison, first, the attack results in Figure 1 consistently show an obvious performance gap between E-TS and plain TS. Specifically, E-TS needs about 5 epochs to find the best corruption pattern, while plain TS needs at least 10 to 25 epochs to find the best corruption pattern, see the blue line of E-TS and the orange line of plain TS. Second, we prove the expected pulling times of those non-competitive arms bound in $O(1)$ compared to $O(\log T)$ in plain TS. It is shown in the fast convergence of E-TS, that discards the exploration of those non-competitive arms thus exploring only the competitive arms and finding the best arm more quickly. For example, we have arms 1,2,3, with the best arm 1, and competitive arms 1 and 2. Using E-TS, we just need to play 1 and 2 multiple times to estimate an accurate prior such that recognizes the best arm 1. However, in plain TS, we also need to play arm 3 multiple times to get more reward information thus wasting the opportunity to use arm 1.
>
> ## Question 10:
>
> **Q:** Figure 2: Fashion MNIST has 7 clients; so even with all 7 clients corrupted, manifold defense does pretty well. This is a surprising and unexpected result.
>
> **R:** The advantage of manifold projection, compared with other defense approaches, can be attributed to its ability to learn the manifold of a class of inputs and recover the manifold to a clean input. While Randomized Smoothing and Dropout can reduce the effect caused by the adversarial part, they cannot completely eliminate the adversarial input and transform it into clean input. In contrast, manifold projection uses an autoencoder to clean the adversarial perturbations before they are fed into the server's model. Our work primarily focuses on selecting the most vulnerable clients in an adaptive attack setting, and we have not specifically designed our approach to overcome the manifold projection defense. Future research may explore methods for moving the input outside the manifold to bypass the manifold projection defense effectively.
>
> ## Question 11:
>
> **Q:** Figures 2, 3 (and parts of 1): Figures showing performance results on Credit Score or the Caltech dataset would have been more meaningful. More specifically, recognition of MNSIT / Fashion MNIST / CIFAR does not require all pieces of the full image; neither does sentiment analysis of a review; further, as noted earlier, such segmentation is unrealistic in any case.
>
> **R:** Thanks for the suggestion. See the response to Q6.
>
> ## Question 12:
> **Q:** Figure 4: What is the setting with 16 and 28 clients? How were the images segmented? Is it meaningful to have a 28-pixel image segment at each of 28 clients?
>
> **R:** We conducted the targeted attack on FashionMNIST with 16 and 28 clients following the setting $\beta = 0.3, Q= 2000, t_0 = 80$, and the test epoch is composed of 25 attack rounds. For the original $28\times28$ data sample, each client has segment $7\times7$ pixels in 16 clients setting and $28\times1$ pixels in 28 clients setting. We do this following previous works (see response to Q6). A more suitable CV dataset in the VFL scenario needs to be investigated.

---

> ### Author Response · Authors · 2023-11-18
> **Response to reviewer XBEx (Part 5/8)**
>
> ## Question 13:
>
> **Q:** Theoretical results Lemma 1, Lemma 2, and Theorem 1 appear to have an implicit large N assumption. The lower bound on the probability is negative for $N < 55$, and exceeds 0.5 at about N=150.
>
> R: The same concern was raised by reviewer HDL1. Having noticed the previous loose bound, we **tighten** the lower bound and update the probability from $\frac{1}{2}\left(1 - \frac{1}{N^2} e^{4\sqrt{\log(N)}}\right)\left(1-e^{-\frac{N}{\sqrt{4\pi N\log(N)}}}\right)$ to $ \left(1-\frac{\delta}{1+ \delta ^2}\phi(\delta)\right) \left(1 - e^{-2(\log(N)-1 )^2}\right)\left(1-e^{-\frac{N}{\sqrt{4\pi N\log(N)}}}\right)= f(\delta)p(N)$ for $\delta > 0, \phi(\delta) = \frac{e^{-\delta^2/2}}{\sqrt{2\pi}}$, where the minimum of $f(\delta)$ is 0.852 and the minimum of $p(N)$ is 0 (increases with $N$). Due to space limitations and complex expressions, we include the detailed proof and figure of the function in the revised Appendix (see **Appendix E and Figure 10**).
>
> In summary, the new bounds provide a positive lower bound for probability and the bound increases with $N$. We draw the function of $ p(N) = \left(1 - e^{-2(\log(N)-1 )^2}\right)\left(1-e^{-\frac{N}{\sqrt{4\pi N\log(N)}}}\right)$ and $ f(\delta) = 1-\frac{\delta}{1+ \delta^2}\phi(\delta) $ and discussed their behaviors in **Appendix E**.
>
> ## Question 14:
>
> **Q:** How does the proposed E-TS algorithm work compared to sequential arm elimination proposed in S. Shahrampour, M. Noshad and V. Tarokh. On Sequential Elimination Algorithms for Best-Arm Identification in Multi-Armed Bandits. IEEE Transactions on Signal Processing, vol. 65, no. 16, pp. 4281-4292, 15 Aug.15, 2017
>
> **R:**
> The referred nonlinear sequential elimination, which we call **NSE** for short, gives a solution to the MAB problem purely by exploring arms. NES is based on the concept of sequentially eliminating arms in an MAB setup until a single best arm is identified. NSE, distinct from traditional sequential elimination methods that linearly allocate exploration resources, employs a nonlinear function relative to the remaining arms for budget allocation in each round. In contrast to E-TS and TS, which focus on minimizing regret, NSE aims to reduce the misidentification probability within a limited exploration budget.
>
> We experimentally compare the performance of the proposed E-TS, plain TS, and NSE. To facilitate a fair comparison, we set different exploration budgets (denoted as $T$) and parameter $p$ for NSE. In this setup, NSE explores each arm's reward for $T$ rounds and subsequently uses the best arm it identifies post-$T$ rounds. We report all experimental results **in Figure 12 and 13 in Appendix G**.
>
> Overall, plain TS has the worst performance. E-TS maintains **stable and faster convergence**, or equivalently, less regret. In contrast, NSE's performance, though sometimes comparable, is **less consistent and requires careful tuning** of hyperparameters $T$ and $p$.
>
> ## Question 15:
>
>
> **Q:** Is a factor 2 missing in the equation for Fact 1?
>
> **R:** Thanks for raising this issue and it is indeed a typo.
> We have added this factor 2 in the revised draft. We note that with this modification, the result in the statement of Lemma 1-5 and Theorem 1 still holds (please refer to proofs for more details), since in all proofs, we just use the **one side of tail bound**, i.e.,
> $
> \operatorname{Pr}\left(\frac{\sum_{i=1}^nX_i}{n} - \mathbb{E}(X_i)\geq \delta\right) \leq \exp{\left(\frac{-2n\delta^2}{(b-a)^2}\right)}
> $
> and
>
> $
> \operatorname{Pr}\left(\frac{\sum_{i=1}^nX_i}{n} - \mathbb{E}(X_i)\leq -\delta\right) \leq \exp{\left(\frac{-2n\delta^2}{(b-a)^2}\right)}, \forall \delta >0.
> $

---

> ### Author Response · Authors · 2023-11-18
> **Response to reviewer XBEx (Part 6/8)**
>
> ## Question 16:
>
> **Q:** In the proof of Lemma 4, shouldn't the = sign be $\leq$ on line 5. The transition from line 6 to line 7 (“g”) is not clear.
>
> **R:** First, we apologize for the typo in line 4 in the proof of Lemma 4. The $\Delta_1$ in line 4 $(\hat{\mu}\_k(t)\geq \mu\_{1}-\frac{\Delta\_{1}}{2})$ should be $\Delta\_k$, which equals to $\mu\_1 - \mu\_k$. We derive $\mu\_1 - \frac{\Delta_k}{2} = \mu_1 - \frac{\mu_1 - \mu_k}{2} = \frac{\mu_1 + \mu_k}{2} = \mu_k +  \frac{\mu_1 - \mu_k}{2} = \mu_k + \frac{\Delta_k}{2}$, thus $(\hat{\mu}\_k(t)\geq \mu\_{1}-\frac{\Delta\_{k}}{2})$ can be transformed to $(\hat{\mu}\_k(t)\geq\mu\_k+\frac{\Delta\_{k}}{2})$ and the equality holds true.
>
> For your convenience, we show the inequality (g) here as follows, termed as(A),
>
> $\sum\_{t=t\_0+1}^T \operatorname{Pr}\left(\hat{\mu}\_{1}(t)-\mu\_{1}\leq-\frac{\Delta_{k}}{2}, n\_{1}(t)\geq \frac{(t-1)}{N}\right)
> +\sum\_{t=t\_0+1}^T \operatorname{Pr}\left(\hat{\mu}\_k(t)-\mu\_k\geq\frac{\Delta\_{k}}{2}, n\_k(t) \geq \frac{(t-1)}{N}\right)$
>
> $\stackrel{(g)}\leq \sum\_{t=t_0+1}^T  2 \exp \left(\frac{-(t-1) \Delta\_{k}^2}{2N}\right)$
>
> To show (g), we start with the first term on the left-hand side of (A):
>
> $\operatorname{Pr}\left(\hat{\mu}\_{1}(t)-\mu\_{1}\leq-\frac{\Delta_{k}}{2}, n\_{1}(t)\geq \frac{(t-1)}{N}\right) \stackrel{(1)}\leq \operatorname{Pr}\left(\hat{\mu}\_{1}(t)-\mu\_{1}\leq-\frac{\Delta_{k}}{2}\bigg | n\_{1}(t)\geq \frac{(t-1)}{N}\right)$
>
> $\stackrel{(2)}= \operatorname{Pr}\left(\frac{\sum_{\tau = 1}^t r_1(\tau)\mathbb{1}(k(\tau) = 1)}{n_1(t)} - \mu_1 \leq - \frac{\Delta_k}{2}| n_{1}(t)\geq \frac{(t-1)}{N}\right)$
>
> $\stackrel{(3)}= \operatorname{Pr}\left(\frac{\sum_{i = 1}^{n_{1}(t)} r^1_i}{n_{1}(t)}- \mu_1 \leq - \frac{\Delta_k}{2}\bigg | n_{1}(t)\geq \frac{(t-1)}{N}\right)$
>
> $\stackrel{(4)}\leq \exp \left(\frac{-n_1(t)\Delta_{k}^2}{2}\right) \leq \exp \left(\frac{-(t-1)\Delta_{k}^2}{2N}\right)$
> where (1) is due to the inequality $\operatorname{Pr}(a\cap b)\leq \operatorname{Pr}(a|b)$, (2) is from the definition of
> $\hat{\mu}\_1(t) = frac{\sum\_{\tau = 1}^t r_1(\tau)\mathbb{1}(k(\tau) = 1)}{n\_1(t)}$.
> In (3), we define $\left(r^1\_i \right)_{i=1}^{n\_1(t)}$
>
> $= \left( r\_1(\tau) \mathbb{1} (k(\tau) = 1);r\_1(\tau)\mathbb{1}(k(\tau) = 1) \neq 0 \right)_{\tau=1}\^t$
>
> which are i.i.d random  with mean $\mu_1$ and bounded support in $[0,1]$. Then  (4) is obtained via Fact 1 (Hoeffding’s inequality) and the condition that $n_{1}(t)\geq \frac{(t-1)}{N}$. Similarly, for the second term on the left-hand side of line 6 in the proof of Lemma 4
>
> , we can derive that $
> \operatorname{Pr}\left(\hat{\mu}\_k(t)-\mu\_k\geq\frac{\Delta\_{k}}{2}, n\_k(t) \geq \frac{(t-1)}{N}\right)\leq \exp \left(\frac{-(t-1)\Delta\_{k}^2}{2N}\right).
> $
> Therefore the inequality (g) holds.

---

> ### Author Response · Authors · 2023-11-18
> **Response to reviewer XBEx (Part 7/8)**
>
> ## Question 17:
>
> **Q:** Proof of Lemma 5: The fact that … towards the end of the proof is not obvious; it holds asymptotically; why does it hold for all T?
>
> Proof of Theorem 1: Please add proof details. The terse statements are insufficient.
>
> **R:** For the puzzling proof of lemma 5, especially in your mentioned part inequality (p) and (q), we rewrite them here as $(B)$ for convenience:
>
> $\sum_{\ell \in[N]\setminus 1}\operatorname{Pr}\left(\left((\hat{\varphi}\_1(t)< \mu_1 - \frac{\Delta\_{min}}{2}) \bigcup (\hat{\mu}\_{\ell}(t) >\mu\_1 - \frac{\Delta\_{min}}{2})\right),n\_1(t)\geq \frac{(t-1)}{N}, n\_{\ell}(t)\geq \frac{(t-1)}{N}\right)$
>
> $+ \sum_{\ell \in[N]\setminus 1}\operatorname{Pr}\left(\left((\hat{\mu}\_1(t)< \mu\_1 - \frac{\Delta\_{min}}{2}) \bigcup (\hat{\mu}\_{\ell}(t) >\mu\_1 - \frac{\Delta\_{min}}{2})\right),n\_1(t)\geq \frac{(t-1)}{N}, n\_{\ell}(t)\geq \frac{(t-1)}{N}\right)$
>
> $\stackrel{(p)}\leq \sum_{\ell \in[N]\setminus 1}\operatorname{Pr}\left((\hat{\varphi}\_1(t)< \frac{\sum_{t=1}^T\mathbb{E}[r^{\max}\_1(t)]}{T} - \frac{\Delta\_{min}}{2},n\_1(t)\geq \frac{(t-1)}{N} \right)$
>
> $+ \sum_{\ell \in[N]\setminus 1}\operatorname{Pr}\left((\hat{\mu}\_1(t)< \mu\_1 - \frac{\Delta_{min}}{2},n\_1(t)\geq \frac{(t-1)}{N} \right)$
>
> $+ 2\sum_{\ell \in[N]\setminus 1}\operatorname{Pr}\left(\hat{\mu}\_{\ell}(t) >\mu\_{\ell} + \frac{\Delta\_{min}}{2}, n\_{\ell}(t)\geq \frac{(t-1)}{N}\right)$
>
> $\stackrel{(q)}\leq 4(N-1)t\exp\left( -\frac{(t-1)\Delta\_{min}^2}{2N}\right),$
>
> To show (p), we start from the first term of line 1 in $(B)$.
>
> $\sum_{\ell \in[N]\setminus 1}\operatorname{Pr}\left(\left((\hat{\varphi}\_1(t)< \mu\_1 - \frac{\Delta_{min}}{2}) \bigcup (\hat{\mu}\_{\ell}(t) >\mu_1 - \frac{\Delta\_{min}}{2})\right),n\_1(t)\geq \frac{(t-1)}{N}, n\_{\ell}(t)\geq \frac{(t-1)}{N}\right)$
>
> $\stackrel{(5)}\leq\sum_{\ell \in[N]\setminus 1}\operatorname{Pr}\left((\hat{\varphi}\_1(t)< \mu_1 - \frac{\Delta\_{min}}{2}),n\_1(t)\geq \frac{(t-1)}{N}, n\_{\ell}(t)\geq \frac{(t-1)}{N}\right)$
>
> $+ \sum_{\ell \in[N]\setminus 1}\operatorname{Pr}\left((\hat{\mu}\_{\ell}(t) >\mu_1 - \frac{\Delta\_{min}}{2}),n_1(t)\geq \frac{(t-1)}{N}, n\_{\ell}(t)\geq \frac{(t-1)}{N}\right)$
>
> $\stackrel{(6)}\leq\sum_{\ell \in[N]\setminus 1}\operatorname{Pr}\left(\hat{\varphi}\_1(t)< \mu_1 - \frac{\Delta\_{min}}{2},n\_1(t)\geq \frac{(t-1)}{N}\right)$
>
> $+ \sum_{\ell \in[N]\setminus 1}\operatorname{Pr}\left(\hat{\mu}\_{\ell}(t) >\mu_1 - \frac{\Delta_{min}}{2}, n\_{\ell}(t)\geq \frac{(t-1)}{N}\right)$
>
> $\stackrel{(7)}\leq \sum_{\ell \in[N]\setminus 1}t\operatorname{Pr}\left(\hat{\varphi}\_1(t)<  \frac{\sum_{t=1}^T\mathbb{E}[r^{\max}\_1(t)]}{T} - \frac{\Delta\_{min}}{2},n\_1(t)\geq \frac{(t-1)}{N}\right)$
>
> $+ \sum_{\ell \in[N]\setminus 1} t\operatorname{Pr}\left(\hat{\mu}\_{\ell}(t) >\mu_{\ell} + \frac{\Delta\_{min}}{2}, n\_{\ell}(t)\geq \frac{(t-1)}{N}\right)$
>
> $\stackrel{(8)}\leq 2(N-1)t\exp\left( -\frac{(t-1)\Delta\_{min}^2}{2N}\right)$
>
>
> where for inequality $(5)$, we simply use the union bound $\operatorname{Pr}(a \cup b)\leq \operatorname{Pr}(a) +\operatorname{Pr}(b)$. For inequality (6), we use the inequality $\operatorname{Pr}(a \cap b \cap c)\leq \operatorname{Pr}(a \cap b) $. As for inequality (7), we first multiply a factor $t$, as $t \geq 1$. Besides, we use the fact that $\frac{\sum_{t=1}^T\mathbb{E}[r^{\max}\_1(t)]}{T} \geq \mu\_1$, where $\frac{\sum_{t=1}^T\mathbb{E}[r^{\max}\_1(t)]}{T}$ is the expected maximum value of arm 1. Since the maximum of arm 1 must be larger than the mean, i.e., $\frac{\sum_{t=1}^T\mathbb{E}[r\_1(t)]}{T}, r\_1(t) \leq r^{\max}\_1(t)$, we substitute $\mu\_1$ with $\frac{\sum_{t=1}^T\mathbb{E}[r\_1(t)]}{T}$. We also use the fact that $\mu\_1 - \frac{\Delta\_{\min}}{2} = \mu\_{k^{\prime}} + \frac{\Delta\_{\min}}{2} \geq \mu\_{\ell} + \frac{\Delta\_{min}}{2}$, since $\mu_{k^{\prime}}$ is arm with second largest mean value and $\mu_{k^{\prime}} \geq \mu_{\ell}, \forall \ell \in [N]\setminus 1$. The last inequality (8) is derived similarly to inequality (1)-(4) as above Q 16 (Part 6) and we sum up all the probability from $\ell \in [N]\setminus 1$. We can also derive the probability bound for the second term of $(B)$ same as the process, thus, the inequality $(p)$ and $(q)$ hold.
>
>  Note that Lemma 5 is **not an asymptotical result** and we have not assumed a value for $t$, the bound holds for any $t\in[T]$.
>
>  Thanks for your reminder. We have added more **details** to the proof of Theorem 1, (see **Appendix B**).

---

> ### Author Response · Authors · 2023-11-18
> **Response to reviewer XBEx (Part 8/8)**
>
> ## Question 18:
> **Q:** Puzzling statement in Section 3: “$\beta \in [0, 1]$ is the perturbation budget of some simple magnitude-based anomaly detector.” Whose anomaly detector, and why is it assumed to be simple?
>
> **R:**  In our threat model, we consider the presence of a simple anomaly detector on the server side. This detector's role is to monitor the **magnitude of embedding changes**, thereby identifying potential adversarial attacks. It operates on the principle that if the deviation in a sample's embedding exceeds a certain threshold (defined by the perturbation budget $\beta$), the sample is flagged as adversarial. This design choice for the anomaly detector is in line with assumptions made in previous works [1] [2] [3], where a perturbation budget is assumed. These studies employ a perturbation budget, assuming the server defending against attacks by evaluating the magnitude of noise rather than employing more advanced techniques like random smoothing or manifold projection.
>
> [1] Ilyas et al. (2018) Black-box Adversarial Attacks with Limited Queries and Information.
>
> [2]Yu et al. (2020) CloudLeak: Large-Scale Deep Learning Models
> Stealing Through Adversarial Examples
>
> [3] Chen et al. (2020) HopSkipJumpAttack: A Query-Efficient
> Decision-Based Attack
>
> ## Question 19:
> **Q:**
> Before (2): “taking over” should be “taken over”.
> P6, line 7: “obtain” should be “obtains”.
>
> **R:** Thanks for your careful reading, we have corrected these typos in the revised draft.
>
> ## Question 20:
> **Q:**
> The sentence about the large exploration space (below Fig 1 and Fig 2) is confusing. How are these numbers obtained?
>
> **R:** The size of the exploration space (or the number of possible corruption patterns) is computed as $N = \binom{M}{C}$. In our experimental settings, there are scenarios with $M=16$ and $28$ clients, and two corruption constraints of $C=2$ and $3$. Therefore, we calculate the size of the exploration space, e.g., $\binom{28}{3} = 3276$.

---

> > ### Comment · Reviewer_XBEx · 2023-11-19
> > **Comments on Authors' Responses**
> >
> > I appreciate the detailed responses to most of my comments, and the 5.5 pages of additional material in the supplement.
> >
> > The statement (and proof) of Lemma 3 has changed. Why. How does it impact the rest of the theoretical development. The author responses do not mention this (although the changed statement is shown in red in the revised supplement).

---

> > > ### Author Response · Authors · 2023-11-19
> > > **Clarifying changes to Lemma 3 and its impact on the rest of the theoretical development**
> > >
> > > Upon careful re-evaluation, we found that our previous probability bound, $\frac{1}{2}\left(1 - \frac{1}{N^2} e^{4\sqrt{\log(N)}}\right)\left(1-e^{-\frac{N}{\sqrt{4\pi N\log(N)}}}\right)$, as noted by you and reviewer HdL1, is relatively loose (e.g., could yield negative values in certain cases).
> > > To address this, we revised the bound to $\left(1-\frac{\delta}{1+ \delta ^2}\phi(\delta)\right) \left(1 - e^{-2(\log(N)-1 )^2}\right)\left(1-e^{-\frac{N}{\sqrt{4\pi N\log(N)}}}\right)$ for $\delta > 0$ and $\phi(\delta) = \frac{e^{-\delta^2/2}}{\sqrt{2\pi}}$. This new bound is **consistently positive and offers a higher probability**, especially when the number of arms $N$ is large (See the plot of probability in Appendix E). It provides a higher confidence level for Lemmas 1, 2, 4, 5, and Theorem 1.
> > >
> > > In the proof of Lemma 3, the overall working flow of deriving this probability bound remains unchanged. We only refined some inequalities via adopting tighter bounds on tail probabilities of Gaussian distributions (refer to the proof of Lemma 3 for details).
> > >
> > > Regarding the impact on the rest of the theoretical development, Lemma 3 establishes the probability bound for the event where the number of times the best arm is pulled does not exceed $(\frac{t-1}{N})$ at round $t$ (defined as $E_1(t)$). The results of Lemma 1, 2, 4, 5, and Theorem 1 are obtained conditioned on that $E_1(t)$ occurs. Therefore, the upper bounds in Lemma 1, 2, 4, 5, and Theorem 1 and their proof processes will not be affected by the change of probability bound in Lemma 3,
> > > e.g., $\mathbb{E}(n_k^{kc}(T))\leq \mathcal{O}(1)$ in Lemma 1, remain unaffected by the change in Lemma 3. The only modification is that the bounds for Lemmas 1, 2, 4, 5, and Theorem 1 are now valid with a higher probability of $E_1(t)$ occurring (the new bound in revised Lemma 3), which reinforces the usefulness of these lemmas and Theorem 1.

---

### Official Review · Reviewer_HdL1 · 2023-11-01

**Soundness:** 1 poor
**Presentation:** 3 good
**Contribution:** 2 fair
**Rating:** 5
**Confidence:** 3

**Summary:**

This paper delves into the vulnerabilities of Vertical Federated Learning (VFL)  through developing a novel attack to disrupt the VFL inference process, under a practical scenario where the adversary can adaptively corrupting a subset of clients. This paper frames this problem as an online optimization problem consisting of adversarial example generation (AEG) and corruption pattern selection (CPS). The paper equates CPS to a multi-armed bandit problem, proposing the Thompson sampling with Empirical maximum reward (E-TS) algorithm, which efficiently identifies the best clients to corrupt, backed by analytical regret bounds and empirical results showcasing its effectiveness.

**Strengths:**

Originality: The paper introduces a novel attack to disrupt the VFL inference process, under a practical scenario where the adversary can adaptively corrupting a subset of clients. This paper frames the attack problem as an online optimization problem consisting of adversarial example generation (AEG) and corruption pattern selection (CPS). CPS is solved by an bandit algorithm which is quite interesting and novel.

Quality: The paper provides a rigorous analysis of the regret bound of the proposed E-TS algorithm. The paper also presents an attack algorithm for attackable instances. The paper also empirically evaluate the performance of the proposed attack on datasets with four major types of VFL tasks.

Clarity: The paper is well-written and  presents the concepts, definitions, and analyses with clarity and precision. The detailed presentation of the experimental setup and outcomes ensures comprehensibility and ease of replication for readers.

Significance: The paper characterize the regret bound of the proposed E-TS algorithm, and empirically demonstrate its capability of efficiently revealing the optimal corruption pattern with the highest attack success rate. The idea of using bandit algorithm to perform optimal attacks may have practical implications for designing efficient attack strategies.

**Weaknesses:**

Overall, I like the formulation of the attack problem and the idea of utilizing bandit algorithm to reveal the optimal corruption pattern. However, I have the following concerns:

1. The regret bound is a probability bound which only holds with some probabilities. However, the probabilities are not in the form like $1-p$. The probabilities depend on N and are less than $\frac{1}{2}$ for any N. If N is smaller than 10, the probabilities are further smaller than 0.07. I doubt whether this regret bound really provide enough information about the effectiveness of the proposed algorithm.

2. In Lemma 2 and Theorem 1, the regret guarantee requires that the selected warm-up rounds $t_0$ satisfies $\sqrt{\frac{t_0}{8N log(t_0)}} \ge \frac{1}{\Delta}$. As the adversary is a black-box adversary, $\Delta$ is unknown to the adversary, and there is no prior information about the choice of the $t_0$. The proper choice of $t_0$ is still a problem.

**Questions:**

1. See in Weakness 1. Can authors explain the choice of the probabilities and the bounds?

2. The performance of the proposed algorithm is related to the choice of $t_0$. The requirement of the prior information of $t_0$ violates the black-box setting. Can authors provide any insight into the proper choice of $t_0$ in the black-box setting?

3. In my opinion, the corruption pattern selection (CPS) problem is more like the setting of combinatorial multi-armed bandits (CMAB) with bandit feedback than MAB. Why did the author choose the model of MAB? Are there some benefits of MAB?

---

> ### Author Response · Authors · 2023-11-18
> **Response to reviewer HdL1 (Part 1/2)**
>
> Thank you to the reviewers for your valuable feedback on our paper. We appreciate the time you spent reviewing our work and your insightful comments. This rebuttal will address the main points you raised. We're eager to clarify these points and improve our paper based on your suggestions.
>
> ## Weakness 1:
>
> **Q:** The regret bound is a probability bound which only holds with some probabilities. However, the probabilities are not in the form like $1-p$
> . The probabilities depend on N and are less than $\frac{1}{2}$ for any N. If N is smaller than 10, the probabilities are further smaller than 0.07. I doubt whether this regret bound really provide enough information about the effectiveness of the proposed algorithm.
>
> **R:** Thank you for highlighting the issue with our probability bound. Upon re-evaluation, we recognized that the initial lower bound was too loose. We have since **tightened** this bound in the revised draft and appendix, (**see Section 6 and Appendix B, E**).
>
>  Specifically, we previously derive the bound $ \frac{1}{2}p $, where $ p = \left(1 - \frac{1}{N^2} e^{4\sqrt{\log(N)}}\right)\left(1-e^{-\frac{N}{\sqrt{4\pi N\log(N)}}}\right) $. The probability $\frac{1}{2}$ is from the probability $ \operatorname{Pr} (\theta_{k'}(\tau_1) \leq \mu_1)p \geq \frac{1}{2}p$. Here, $ \theta_{k'}(\tau_1) \leq \mu_1 $ indicates that the sample from a suboptimal arm is less than the mean of the best arm. Assuming a Gaussian distribution for the suboptimal arm with mean $ \mu_{k'} $ and variance  $\sigma_{k'}^2 $, and given $ \mu_{k'} \leq \mu_1 $, we derive the lower bound of the probability $ \operatorname{Pr} (\theta_{k'}(\tau_1) \leq \mu_1) \geq \frac{1}{2} $.
>
> To tighten this bound further, let $ x = \theta_{k'}(\tau_1) - \mu_1 $ and normalize $ x $ as $ \tilde{x} \sim \mathcal{N}(0,1)$:
>
> $
>  \operatorname{Pr} (\theta_{k'}(\tau_1) \leq \mu_1) = \operatorname{Pr} (x \leq 0) = \operatorname{Pr}(\tilde{x} \leq \frac{\mu_1 - \mu_{k'}}{\sigma_{k'}})$
>  $
>  =1 - Q\left(\frac{\mu_1 - \mu_{k'}}{\sigma_{k'}}\right) \geq 1-\frac{\delta}{1+ \delta^2}\phi(\delta),
> $
>
> where $ Q(\cdot) $ is the Q function as the tail probability of a standard normal distribution~[1], $ \delta = \frac{\mu_1 - \mu_{k'}}{\sigma_{k'}} $, and $ \phi(\cdot) $ is the density function of the standard normal distribution.
> With this tighter bound, the lower bound improves from $ 0.5p $ to $ (1-\frac{\delta}{1+ \delta^2}\phi(\delta))p $, which is at least $ 0.852p $, (see plot of the function **in Figure 10 (a) of Appendix E**).
>
> Regarding $ p $, we initially used a looser bound, leading to negative values in some cases. We have now adopted a tighter bound from [2], defined as $ p = \left(1 - e^{-2(\log(N)-1 )^2}\right)\left(1-e^{-\frac{N}{\sqrt{4\pi N\log(N)}}}\right) $, which minimum is 0 and increase with greater $N$. Due to space limitations and complex expressions, we include the detailed proof and figure of the function in the revised appendix, (see **Appendix E Figure 10 (b)**).
>
> In summary, the new bounds provide a **positive** lower bound for probability and the probability **increases** with greater $N$. We draw the function of $ p(N) = \left(1 - e^{-2(\log(N)-1 )^2}\right)\left(1-e^{-\frac{N}{\sqrt{4\pi N\log(N)}}}\right)$ and $ f(\delta) = 1-\frac{\delta}{1+ \delta^2}\phi(\delta) $ and discussed its trend in the Appendix E.
>
> Typically, $ N $ is large in practical scenarios due to its combinatorial nature, even for relatively small number of of clients, and E-TS is designed to manage this extensive exploration space. For instance, the number of arms $ N $ is at least 120, implying $ p(N) \geq 0.756 $. For smaller $N $ values, such as $ N = 21 $ with $ p = 0.52 $ in the FashionMNIST attack (Figure 1 (a)), the lower probability bound may seem modest, yet the empirical performance demonstrates rapid convergence compared to plain TS. Overall, our E-TS algorithm consistently shows faster convergence than TS in practice, and we can theoretically guarantee a high probability bound for large $ N $.
>
> [1] Wikipedia Q-function https://en.wikipedia.org/wiki/Q-function
>
> [2] AGRAWAL et al. (2017) Near-Optimal Regret Bounds for Thompson Sampling

---

> ### Author Response · Authors · 2023-11-18
> **Response to reviewer HdL1 (Part 2/2)**
>
> ## Weakness 2:
>
> **Q:** In Lemma 2 and Theorem 1, regret guarantee requires that the selected warm-up rounds $t_0$
>  satisfies $\sqrt{\frac{t_0}{8N\log t_0}}\geq \frac{1}{\Delta}$. As the adversary is a black-box adversary, $\Delta$ is unknown to the adversary, and there is no prior information about the choice of the $t_0$. The proper choice of
> $t_0$ is still a problem.
>
> **R:** Regrading your concern, we have added related discussion in **Appendix C.4**.
>
> Our theoretical framework presumes a specific $t_0$ value. In practical scenarios, setting $t_0$ to be at least larger than the number of arms $N$ is usually sufficient. Our experiments (**Figure 7 (a) and (b) in Appendix C.4**) indicate that a range between
> $2N $ and $5N$ is optimal. This range ensures that each arm is sampled at least twice using TS, providing the adversary with a more accurate initial understanding of each arm's prior distribution. This preliminary knowledge is crucial for the E-TS to effectively identify an empirical competitive set of arms. If $t_0$ is too small, there's a risk of E-TS prematurely converging on a suboptimal arm due to insufficient initial data, potentially omitting the optimal arm from the empirical competitive set.
>
> The optimal range of $t_0$ can be further validated in Figure 3 (b) in Section 7.2, where the number of arms $N =21$ and the optimal range of $t_0$ should be $[42, 105]$. From figure 3(b), we can observe that if we set $t_0 = 20$, E-TS converges to a suboptimal arm, and when $t_0 = 80$ and $t_0 = 200$, E-TS can converge to the optimal arm. Specifically, E-TS with $t_0 = 80$ has faster convergence than the one with $t_0 = 200$ because, within 80 rounds, E-TS has enough information of each arm to construct the empirical competitive set, thus discarding the non-competitive arm.
>
>
> In sum, the optimal range of $t_0$ is $[2N, 5N]$ in practice, aiding in the accurate estimation of arm distributions and enhancing the overall effectiveness of E-TS.
>
>
> ## Question 1:
>
>
> **Q:** See in Weakness 1. Can authors explain the choice of the probabilities and the bounds?
>
> **R:** See response 1 and Appendix B and E.
>
> ## Qustion 2:
> **Q:** The performance of the proposed algorithm is related to the choice of $t_0$
> . The requirement of the prior information of $t_0$ violates the black-box setting. Can authors provide any insight into the proper choice of $t_0$ in the black-box setting?
>
>  **R:** Please refer to response 2.
>
> ## Question 3:
> **Q:** In my opinion, the corruption pattern selection (CPS) problem is more like the setting of combinatorial multi-armed bandits (CMAB) with bandit feedback than MAB. Why did the author choose the model of MAB? Are there some benefits of MAB?
>
> **R:**
>  We initially considered CMAB for the corruption pattern selection issue but ultimately chose MAB due to several reasons:
>
> 1. **Clear Definition of Rewards:** In a CMAB framework, defining the reward for each base arm is challenging, especially under varying corruption constraints. For instance, using the average ASR (Attack Success Rate) of including this arm to represent a base arm's reward can result in inconsistencies across different constraints. MAB, on the other hand, allows for a straightforward definition of each arm's reward as the ASR when that particular arm (corruption pattern) is used.
>
> 2. **Simplified Assumptions:** CMAB typically assumes the rewards of base arms to be independent and identically distributed (i.i.d.), which does not hold in our context due to the influence of corruption constraints. Moreover, common CMAB models [3] [4] assume that the super arm's reward is a simple aggregate (average or sum) of the base arms' rewards, which does not align with our attack model. Although some CMAB variants [5] [6] consider nonlinear relationships between super and base arms, they still rely on observable rewards for base arms or presume i.i.d. distributions, which are not applicable in our case.
>
> 3. **Comparative Performance and Regret Bound:** The regret bound in most CMAB works is $\mathcal{O}(\log(T))$ [3] [4] [5] [6], similar to MAB solutions. However, our MAB approach improves the regret of non-competitive arms from $\mathcal{O}(\log(T))$ to
> $\mathcal{O}(\log(1))$, showing a significant advantage. Additionally, our initial trials with a standard CMAB algorithm [3] yielded poorer results compared to a plain Thompson Sampling (TS) approach in MAB.
>
> Therefore, considering these aspects, MAB was chosen for its clearer reward structure, less restrictive assumptions, and superior performance in our specific scenario.
>
>
>
> [3] Wang et al. (2018) Thompson Sampling for Combinatorial Semi-Bandits
>
> [4] Combe et al. (2015) Combinatorial Bandits Revisited
>
> [5] Chen et al. (2016) Combinatorial Multi-Armed Bandit with General
> Reward Functions
>
> [6] Perrault et al. (2020) Statistical Efficiency of Thompson Sampling for
> Combinatorial Semi-Bandits

---

> ### Author Response · Authors · 2023-11-20
> **Can you please response to our rebuttal for Paper5036**
>
> Dear Reviewer HdL1,
>
> We are approaching the end of the discussion phase, and we have unfortunately received no feedback from you on our rebuttal.
>
> Please can we kindly ask you to take a look at our responses, and let us know whether we have clarified your questions and addressed your concerns?
>
> Thank you very much again for the time you spent reviewing.
>
> Paper5036 Authors

---

> > ### Comment · Reviewer_HdL1 · 2023-11-21
> >
> > Thank you for the detailed response. Most of my concerns are addressed. However, I am still not satisfied with the high-probability bounds. It would be better that the bounds hold on probabilities that does not depend on $N$, like just $1-\delta$. The current results imply that the proposed algorithm works only on large $N$ but large $N$ will impact the expected regret of the E-TS algorithm.
> >
> > If we want to make sure that $p(N) \ge (1 - \frac{1}{T^\alpha})$, we need to have $ 1 - e^{-2(\log(N)-1)^2}\ge 1 -  \frac{1}{T^\alpha}$ and $N \ge {T}^{\alpha/2}$. As it suggests that we choose $t_0$ between $2N$ to $5N$, the high-probability regret scales on at least ${T}^{\alpha/2}$ and the expected regret scales on $T^{1-\alpha}+{T}^{\alpha/2}$ with minimum at $\alpha = 2/3$. In fact, the expected regret at lease scales on $T^{1/3}$. The claim of $log(T)$ regret is not convincing,
> >
> > Thus, I will keep my score.

---

> > > ### Author Response · Authors · 2023-11-22
> > > **Response to reviewer HdL1**
> > >
> > > We appreciate the opportunity to further clarify the nature of our regret bound.
> > >
> > > We'd like to clarify that our regret bound is a conditional expected regret bound, rather than a high-probability bound desired by the reviewer.
> > > Specifically, our bound is an upper bound on the expected regret when event $E_1(t)$ happens. As stated in Theorem 1, with the probability $E_1(t)$ happens, our expected regret bound $D\mathcal{O}(\log(T)) + (N-D)\mathcal{O}(1)$ holds.

---

### Official Review · Reviewer_GNem · 2023-11-03

**Soundness:** 2 fair
**Presentation:** 3 good
**Contribution:** 3 good
**Rating:** 6
**Confidence:** 4

**Summary:**

The paper studies the adversarial attack in vertical federated learning inference phase. The adversary is the man-in-the-middle that can only perturbs the intermediate embeddings of the test samples sent by the clients. The authors propose an attack utilizing multi-armed bandit to dynamically choose the best subset of the clients to perturb. The goal of the adversary can be targeted (i.e., the prediction of the test sample is a specific label) and untargeted (i.e., the prediction of the test sample can be any label except a specific one). An ablation experimental study is provided to study how the different parameters (e.g., the number of corrupted channels, perturbation budget) impact the attack.

**Strengths:**

The paper is well-written, not hard to follow. The algorithm is clearly presented. The theoretical part seems solid with classical analysis.

**Weaknesses:**

However, I have some concerns on the attack design and the feasibility of the proposed method (detailed in the questions). I might misunderstand some part, but if the authors can answer my concerns, I would like to adjust my score.   For the correct stage, I would not go for the acceptance.

**Questions:**

Questions:
1) For the VFL inference setting, one point is not clear for me: the server broadcasts the probabilities vector back to all the clients for the inference. However, it would be more cost-saving to broadcast only the label. In this scenario, it would be harder to design attacks. Is there any reason that the client must receive the probability vector for the inference?

2) I would like to see the discussion on how easy/difficult this attack can be noticed by the server. For me, it seems that targeted attack could be easily identified by the server. If the server observes that for test sample queries, the probability vector evolution converges to the same label, i.e., there might be an attack. Then one naïve way to mitigate this attack is that the server sends back always the probability vector obtained by using the first embedding query to avoid the training of $\eta_i^t$. Actually, I do not see the interest for the server to answer $n$ queries of the same test sample (line 10 in Algo 2) which helps the attacker to find the best perturbation.

3) From my intuition, untargeted attack should be easier than the targeted one. If one can succeed the targeted attack with ASR $r$ on label $x$, then one would expect the untargeted one should be at least $r$ on label $y != x$. More precisely, let $r(x)$ be the targeted ASR on label $x$, the untargeted one on label $y$ should be better than $max$ {$r(x)|\forall x!=y$}. In the experiment, the results on FashionMNIST confirms this intuition, whileas the results on CIFAR-10 gives opposite observation. Could authors provide more explanation for this observation? Besides, what are the untargeted labels used in the experiments?

4) In Figure 3c, when the query budget decreases to 500 for FashionMNIST, the targeted attack accuracy reaches almost 10%, which is equivalent to the ASR of no-attack case if the sample is randomly split in the batch.  The same observation can be found in Figure 5c when there is only one corrupted channel. Does it mean that when the query budget is limited or the number of corrupted channels is limited, applying the proposed method has smaller or even no advantage than doing nothing for targeted attack? What impacts the minimum necessary query budget and the minimum required number of corrupted channels? The complexity of the dataset, model or the scale of the system? I would like to have more insight on this problem.

Minors:

1) The settings for each Figure are not well-indicated, especially for the ablation study. For example, in Figure 3, when we vary one parameter, what are the default values for the other parameters.

---

> ### Author Response · Authors · 2023-11-18
> **Response to reviewer GNem (Part 1/3)**
>
> Thank you to the reviewers for your valuable feedback on our paper. We appreciate the time you spent reviewing our work and your insightful comments. This rebuttal will address the main points you raised. We're eager to clarify these points and improve our paper based on your suggestions.
>
> ## Question 1:
> **Q:** For the VFL inference setting, one point is not clear for me: the server broadcasts the probabilities vector back to all the clients for the inference. However, it would be more cost-saving to broadcast only the label. In this scenario, it would be harder to design attacks. Is there any reason that the client must receive the probability vector for the inference?
>
> **R:**  In line with established research on attacks in the API/remote server model [1] [2] [3] [4], queries to an API or remote ML model typically yield probability vectors, the prediction's probability, or just hard labels. Our threat model assumes that the server, functioning as a **remote model**, broadcasts the probability vector to all querying clients. In our attack design, as detailed in equation 3, section 5, these probability vectors are crucial for calculating the loss in the Inner problem AEG. Both targeted and untargeted attacks utilize the **cross-entropy** between the probability vector and the target label (positive loss for targeted attack and negative loss for untargeted attack). It's important to note that if the server were to provide only hard labels or the prediction's probability, our problem formulation and E-TS solution would remain **unaffected**. Alternative loss functions or simply computing the loss between the probability with the target label could be employed to address the inner problem in such scenarios.
>
> ## Question2:
>
> **Q:** I would like to see the discussion on how easy/difficult this attack can be noticed by the server. For me, it seems that targeted attack could be easily identified by the server. If the server observes that for test sample queries, the probability vector evolution converges to the same label, i.e., there might be an attack. Then one naïve way to mitigate this attack is that the server sends back always the probability vector obtained by using the first embedding query to avoid the training of
> . Actually, I do not see the interest for the server to answer
>  queries of the same test sample (line 10 in Algo 2) which helps the attacker to find the best perturbation.
>
>  **R:** The defense method you suggest is indeed practical and represents a potential area for future research. In response to such a defense, an adversary could set different target labels for each sample, preventing the server from detecting label convergence in a batch of data (note that our targeted attack does not require consistent target labels for different test samples). Additionally, even if the adversary uses the same target labels, the initial query results can reveal valuable information about the perturbation direction.
>  This information is useful before the server recognizes and counters the attack, e.g., we conduct the targeted attack in FashionMNIST with target label 6, query budget 500, and $\beta = 0.3$. It can achieve over 50\% ASR with an average 203 queries in two corrupted client setting, which is far lower than the 500 query budget and indicates that initial direction has a huge effect on the attack.
>
> Regarding the query capacity, our model aligns with previous works on remote server attacks [2] [3] [4]. For instance, Google's Cloud Vision API currently **permits up to 1,800 requests** for a single sample. Our threat model sets specific query ($Q$) and perturbation ($\beta$) budgets for the server, implying that queries remain safe as long as they stay within these limits. If the query budget is limited, the adversary can generate multiple noisy copies of the test samples and set them with different sample IDs in advance to increase the query times stealthily without exceeding the budget.
>
> In summary, straightforward server strategies like adjusting the query budget or monitoring for label convergence can be circumvented. Attackers can use tactics such as **duplicating samples, varying target labels, or employing initial gradient directions** for attack. These methods help in evading detection and effectively conducting attacks.
>
> [1]Yu et al. (2020) CloudLeak: Large-Scale Deep Learning Models
> Stealing Through Adversarial Examples
>
> [2] Chen et al. (2020) HopSkipJumpAttack: A Query-Efficient
> Decision-Based Attack
>
> [3] Chen et al. (2017) ZOO: Zeroth Order Optimization Based Black-box Attacks to
> Deep Neural Networks without Training Substitute Models
>
> [4] Ilyas et al. (2018) Black-box Adversarial Attacks with Limited Queries and Information.

---

> ### Author Response · Authors · 2023-11-18
> **Response to reviewer GNem (Part 2/3)**
>
> ## Question 3:
> **Q:** From my intuition, untargeted attack should be easier than the targeted one. If one can succeed the targeted attack with ASR $r$ on label $x$, then one would expect the untargeted one should be at least $r$ on label
> $y!=x$. More precisely, let $r(x)$
>  be the targeted ASR on label $y$
> , the untargeted one on label
>  should be better than
>  $\max \{r(x)|\forall{x}\neq y\}$. In the experiment, the results on FashionMNIST confirms this intuition, whileas the results on CIFAR-10 gives opposite observation. Could authors provide more explanation for this observation? Besides, what are the untargeted labels used in the experiments?
>
> **R:** Thank you for your insights. In our experiments, the untargeted attacks were designed to deviate from the true labels of the test set. Specifically, if the true label is $y$, the goal of the attack was to ensure the predicted label $\hat{y}$ is not equal to $y$. For targeted attacks on CIFAR-10, we consistently chose the target label as 7. This approach results in differing numbers of attacked samples between targeted and untargeted attacks.
>
> The observed inferior performance in the untargeted attack on CIFAR-10 is primarily due to a lower perturbation budget $\beta = 0.05$ compared to the budget $\beta = 0.08$ used in the targeted attack. To address this, we have re-implemented the untargeted attack with the same perturbation budget of $\beta = 0.08$, and the results have been updated in Figure 6 of **Appendix C.3**. It shows that the untargeted attack has **higher ASR** when setting the same perturbation budget and corruption constraint.
>
> ## Question 4:
> **Q:** In Figure 3c, when the query budget decreases to 500 for FashionMNIST, the targeted attack accuracy reaches almost 10\%, which is equivalent to the ASR of no-attack case if the sample is randomly split in the batch. The same observation can be found in Figure 5c when there is only one corrupted channel. Does it mean that when the query budget is limited or the number of corrupted channels is limited, applying the proposed method has smaller or even no advantage than doing nothing for targeted attack? What impacts the minimum necessary query budget and the minimum required number of corrupted channels? The complexity of the dataset, model or the scale of the system? I would like to have more insight on this problem.
>
> **R:**
> The experimental results in Figure 3 (a) indeed show that with only one corrupted channel and $Q = 2000$, the ASR can exceed 30\%. However, the lower ASR observed when the query budget is under 500 is primarily attributed to the challenges of a targeted attack, which necessitates classifying all examples into a single target, such as label 7. Achieving this projection to label 7 requires extensive gradient estimation, whereas an untargeted attack can nearly reach a 50\% ASR within the same query budget (see Figure 5 (b) in Appendix C.2).
>
> Furthermore, **the ASR of a targeted attack is influenced by the choice of the target**. For instance, when conducting a targeted attack with the target label set to 6, under the same conditions (a query budget of 500 and a corruption budget of 2), the E-TS best arm results indicate an ASR of 51\%. Remarkably, even with the same query budget but only one corrupted channel, the ASR can reach 37\% with an average of 203 queries.
>
> In conclusion, our proposed attack method can be effective even with a single corrupted channel and limited query budget. However, the attack's performance is significantly impacted by factors such as the type of attack (targeted or untargeted) and the specific target label chosen.
>
> To explore how the necessary number of queries and corrupted channels vary across different models,
> datasets, and systems, we conducted experiments using Credit and Real-sim datasets. We specifically
> analyzed the average number of queries required to attain a 50\% ASR under various levels of client
> corruption (corruption constraint $C$). The detailed results and discussion are in **Appendix D**.
>
> From the results in **Figure 9 (a) and (b) in Appendix D**, to attain a target ASR with the same $C$, simpler datasets within the same task require
> fewer queries. Systems with a higher number of clients necessitate more queries. However, the
> influence of the model’s complexity on the number of queries is ambiguous and warrants further
> investigation.

---

> ### Author Response · Authors · 2023-11-18
> **Response to reviewer GNem (Part 3/3)**
>
> ## Question 5:
> **Q:** The settings for each Figure are not well-indicated, especially for the ablation study. For example, in Figure 3, when we vary one parameter, what are the default values for the other parameters.
>
> **R:** We apologize for any confusion caused by the initial presentation. In the revised draft, we have provided detailed specifications for each figure, particularly concerning the ablation study settings. For the ablation study, the default corruption constraint $C$ is set to 2, the number of warm-up rounds $t_0$ is 80, the query budget $Q$ is 2000, and the perturbation budget $\beta$ is 0.3. These details have been thoroughly updated in both the draft and the appendix.

---

> ### Author Response · Authors · 2023-11-20
> **Can you please response to our rebuttal for Paper5036**
>
> Dear Reviewer GNem,
>
> We are approaching the end of the discussion phase, and we have unfortunately received no feedback from you on our rebuttal.
>
> Please can we kindly ask you to take a look at our responses, and let us know whether we have clarified your questions and addressed your concerns?
>
> Thank you very much again for the time you spent reviewing.
>
> Paper5036 Authors

---

> ### Comment · Reviewer_GNem · 2023-11-20
> **about additional experiment in Appendix D**
>
> First, I appreciate the devoted time and work spent by the authors for the response.
> However, for the additional experiments in Appendix D, I still have some concerns.
> 1) If I understand correctly, these two datasets are binary classifications. So if a batch has half of the data points 0 and half of the data points 1, I would expect that even without attack, the sucess rate would reach 50% when the model predicts correctly according to the definition of the metric. However, the proposed attack requires still >100 queries when 2 clients are corrupted. Otherwise, the metric here is that 50% of the samples have the opposite label of the true one? Or in the batch, there are all samples of 1/0?
>
> 2) For the model complexity, it might be that a model which generalizes better is more robust to the adversarial attack. Is it possible the deeper model in the experiment overfits the real-sim data?

---

> > ### Author Response · Authors · 2023-11-21
> > **Response to reviewer GNem for additional experiment in Appendix D**
> >
> > ## Q1
> > Regarding your first concern, we appreciate the opportunity to clarify. The metric employed here indeed **indicates that 50\% of the samples have predicted labels opposite to their true values**. Specifically, for samples originally (without attack) classified to their true labels, which are either $0$ or $1$, our attack aims to alter the prediction to the opposite value. Consequently, an Attack Success Rate (ASR) exceeding 50\% implies that more than half of the samples are misclassified with labels contrary to their actual ones, and hence a test accuracy of lower than 50\%. Note that this is not the case without attack, for which the VFL model achieves 96.1\% test accuracy.
> >
> > ## Q2
> >
> > We concur that generally, a model demonstrating superior generalization tends to exhibit increased robustness against adversarial attacks. We'd like to point out that the saved models (both Real-sim with deeper layer and shallow layer) in our experiments are **not overfitting**. Specifically, we saved the model at a point where test accuracy plateaued while training accuracy continued to rise. Besides, in our experiments in Appendix D, a deeper model on Real-sim marginally outperforms the original Real-sim model, achieving a test accuracy of 97\% compared with 96.1\%.
> >
> > We conjecture that the lower average number of queries required in a deeper Real-sim model is caused by the **Dropout layer in the model**.
> > Specifically, the standard 3-layer Real-sim model incorporates a Dropout layer after the first layer of the server model. In contrast, the deeper Real-sim model used in our experiment extends the standard version with three additional Linear-ReLU layers following the server model's Dropout layer, but has no Dropout layer as a regularization in the extended layers. This architectural difference is likely to lead the deeper model to learn a **distinct** feature space, potentially rendering it more susceptible to adversarial attacks.
> >
> > To further explore this hypothesis, we introduced an additional model variant, **Real-sim(dropout)**, mirroring the deeper Real-sim model but with an extra Dropout layer located before the penultimate layer of the server model. Both the Real-sim(deep) and Real-sim(dropout) models achieved a test accuracy of 97\%. We then examined the average number of queries required by Real-sim(dropout) under various corruption constraints, with these findings now **included in Figure 9 (a) of Appendix D**.
> >
> > Our results reveal that Real-sim(dropout) requires a higher number of queries to reach 50\% ASR, compared with both Real-sim(deep) and Real-sim(standard), for all numbers of corrupted clients. Dropout can enhance robustness by preventing the model inference from being overly dependent on any single feature or input node, encouraging the model to learn more redundant and robust representations. This outcome supports our conjecture that the adoption of Dropout layers improves model robustness.

---

> ### Comment · Reviewer_GNem · 2023-11-21
>
> Thanks for the clarification. I was confused by the metric defined in the paper. It may need to be redefined in a more proper way.
> I do not have any more concern/question related to the paper.

---

> ### Author Response · Authors · 2023-11-21
> **Response to reviewer GNem**
>
> We are grateful for your feedback and appreciate the increase in your score.
>
> Regarding the metric's clarity, we understand the confusion and would like to offer further explanation, revisiting some aspects of the threat model outlined in our draft.
>
> 1. For the targeted attack, the adversary aims to make the top model output its intended label $\hat{y} = y_v$, for some target label $y_v$. For the untargeted attack, the adversary would like the predicted result $\hat{y}$ to be different from some label $y_u$.
>
> 2. The attack's effectiveness is measured by attack success rate (ASR), which is defined as $ASR^s\_v =  \frac{\sum_{i=1}^s \mathbb{1}(\hat{y}(\boldsymbol{p}\_i) = y_v)}{s}$ and $ASR_u^s = \frac{\sum_{i=1}^s \mathbb{1}(\hat{y}(\boldsymbol{p}\_i) \neq y\_u)}{s}$ for targeted and untargeted attack, respectively, where $s$ is the number of test samples, $\boldsymbol{p}_i$ is the probability vector of test sample $i$, and $\mathbb{1}(\cdot)$ is the indicator function.
>
> Let us specify the metric used in our experiment for untargeted and targeted attack.
>
> There are $N$ test samples, each with sample ID $i, i\in[N]$. We use $[N]$ to denote $[1,\ldots, N]$ and $\hat{y}\_i$ to denote the model's output label for sample $i$ (without attack).
>
> -   **Targeted Attack:** We designate a specific label (e.g., label 7) as the target label $y\_v$. Samples to be attacked are those not initially classified as $y_v$ before attack. Thus, the count of samples to be attacked is $s = \sum_{i = 1}^N \mathbb{1}\left(\hat{y}\_i \neq y_v\right)$. After the attack, we calculate the number of successfully manipulated samples $A_t=\sum_{j\in[s]}\mathbb{1}(\hat{y}\_j = y_v)$ and determine the ASR as $ASR^s_v = \frac{A\_t}{s}$.
>
> -   **Untargeted Attack:** Here, %the true label serves as the target
> for some target label $y_u$, samples to be attacked are those initially classified as $y\_u$ before attack. Therefore, $s = \sum\_{i = 1}^N \mathbb{1}\left(\hat{y}\_i = y\_u\right)$. Note that for a model with high test accuracy, $y_u$ is most of the time the true label of the test sample. After attack, the number of successfully manipulated samples is $A\_u=\sum_{j\in[s]}\mathbb{1}(\hat{y}\_j \neq y\_u)$, leading to the ASR: $ASR^s\_v = \frac{A\_u}{s}$. Note that all attacks on binary classification tasks in our study are treated as untargeted attacks.
>
> We believe the confusion may stem from the statement "s is the number of test samples" in Section 3's metric definition, which could lead to the misconception that all test samples are subject to attack. To clarify, we have modified **Goal of the adversary** part in our Threat Model section to define the samples to be attacked in targeted and untargeted attacks, and revised the statement in the metric definition to be "s is the number of samples to be attacked ".

---

### Author Response · Authors · 2023-11-19
**Global response (Part 1/3)**

We would like to express our sincere gratitude to all the reviewers for dedicating their time and expertise to evaluate our paper. The comprehensive feedback provided has been invaluable in guiding improvements and clarifications in our work. In this global response, we will address the common points and concerns raised by the reviewers, illustrating how each has been thoughtfully considered and incorporated into our revised draft and appendix. Besides, we explicitly list all the modifications of the draft and the appendix. We remain committed to refining our work to meet the high standards of the conference and deeply appreciate the guidance offered through your reviews.

1.  The primary concern raised by reviewers HdL1, XBEx, and qKnn pertains to **the selection of warm-up rounds $t_0$** in our E-TS algorithm.

To address this, we have included additional discussion and experimental results in **Appendix C.4**. Practically, setting $t_0$ to a value greater than the number of arms $N$ is generally adequate. Our experiments, illustrated in Figure 7 of Appendix C.4, demonstrate that a range between $2N$ to $5N$ is optimal. This range ensures that each arm is sampled at least twice through Thompson Sampling (TS), allowing the algorithm to gain a more accurate initial understanding of each arm's prior distribution. This initial assessment is crucial for E-TS to effectively construct an empirical competitive set of arms. If $t_0$ is set too low, E-TS runs the risk of prematurely converging on a suboptimal arm due to a lack of sufficient initial data, potentially excluding the optimal arm from the empirical competitive set.

The efficacy of this optimal range for $t_0$ is further substantiated by Figure 3(b) in Section 7.2. Here, with the number of arms $N = 21 $ the optimal range for $t_0$ is identified as $[42, 105]$. Observations indicate that setting $t_0 = 20$ leads E-TS to converge on a suboptimal arm, whereas $t_0 =80$ and $t_0 = 200$ allow E-TS to identify the optimal arm. Notably, E-TS with $t_0 = 80$ achieves faster convergence than with $t_0 =200$, as 80 rounds provide sufficient information for E-TS to construct the empirical competitive set, thereby disregarding non-competitive arms.

In conclusion, we recommend **an optimal range for $t_0$ between $2N$ and $5N$**. This range significantly contributes to the precise estimation of arm distributions, thereby enhancing the effectiveness of the E-TS algorithm.


2. The second common concern, raised by reviewers GNem and XBEx, pertains to the server's capability in VFL inference, specifically regarding the necessity of **broadcasting probability vectors for predictions** upon receiving a query, as opposed to broadcasting a class with a probability value or simply a hard label.

For this concern, we clarify that our assumption about the server's capability aligns with established research in attack methodologies. Previous works focusing on attacking a remote server or API [1] [2] [3] [4] generally presume that the server provides probability vectors, the prediction's probability, or merely hard labels in response to queries on test samples. In our research, we also assume the server to be a remote model during inference. Our proposed attack is predicated on the scenario where the server broadcasts probability vectors. These vectors are crucial for computing the loss in our work's inner problem, namely Adversarial Example Generation (AEG), specifically evaluating the cross-entropy between the probability vector and the target label.

However, even if the server only broadcasts the probability value of the prediction or hard label, it will not affect the problem formulation and our solution Thompson sampling with Empirical maximum reward (E-TS) to select the best corruption pattern.
 Alternative loss functions for scenarios with hard labels and probability values, such as computing the cross-entropy between the probability value and the target label, can be employed to address the inner problem in these circumstances.

[1]Yu et al. (2020) CloudLeak: Large-Scale Deep Learning Models Stealing Through Adversarial Examples

[2] Chen et al. (2020) HopSkipJumpAttack: A Query-Efficient Decision-Based Attack

[3] Chen et al. (2017) ZOO: Zeroth Order Optimization Based Black-box Attacks to Deep Neural Networks without Training Substitute Models

[4] Ilyas et al. (2018) Black-box Adversarial Attacks with Limited Queries and Information.

---

> ### Author Response · Authors · 2023-11-19
> **Global response (Part 2/3)**
>
> 3. The third common concern is how stealthy our attack is, especially in the **large query budget setting** of our experiments. This concern is from reviewer GNem and XBEx, who worry that the number of queries $Q$ is large, i.e., we set $Q = 2000$ in the experiment, making it easy to detect an attack by the server.
>
> First, we clarify that our assumption aligns with previous works [1] [2] [3] [4], where the server serves as a remote model/API and responds to multiple queries for test samples. For example, the Google Cloud Vision API permits a query budget of up to 1800 for a single sample. Our primary focus is on the efficiency of the E-TS algorithm in selecting the optimal corruption pattern. The inner Adversarial Example Generation (AEG) problem is more akin to traditional adversarial example generation techniques.
>
> Addressing the concern about the large query budget, we conduct a targeted attack on FashionMNIST with the target label set to 6, under the setting that a query budget of 500, perturbation budget of 0.3, and a corruption constraint of 2, the E-TS best arm results show an ASR of 51\% with average 203 queries (compare to the result less than 10\% ASR in target label 7). Furthermore, studies in Appendix D demonstrate that an adversary can successfully attack over 50\% of test samples within 600 queries, even when corrupting only one client channel in a tabular dataset (refer to Figure 9 in Appendix D). This suggests that even with a smaller query budget, a successful attack on half of the samples is feasible in certain scenarios with different target labels and datasets. Secondly, adversaries could duplicate multiple noisy versions of the samples before inference, effectively increasing their query budget more stealthily. Third, recent advancements in query-efficient methods [2] [5] and transfer-based approaches [6] offer alternatives to the AEG problem that do not compromise the efficacy of the E-TS solution.
>
> In conclusion, by employing strategies such as choosing appropriate target labels, duplicating samples in advance, and utilizing query-efficient techniques, the limitations of a restricted query budget can be effectively mitigated.
>
>
> 4. The fourth common concern, raised by reviewers HdL1 and XBEx, revolves around Lemmas 1, 2, and Theorem 1 being conditioned on a **probability bound**. This bound becomes small when the number of arms $N $is small, rendering Lemmas and Theorem 1 less convincing and effective.
>
>
> Upon careful re-evaluation, we found that our previous probability bound, $\frac{1}{2}\left(1 - \frac{1}{N^2} e^{4\sqrt{\log(N)}}\right)\left(1-e^{-\frac{N}{\sqrt{4\pi N\log(N)}}}\right)$ is relatively loose (e.g., could yield negative values in certain cases). To address this, we revised the bound to $\left(1-\frac{\delta}{1+ \delta ^2}\phi(\delta)\right) \left(1 - e^{-2(\log(N)-1 )^2}\right)\left(1-e^{-\frac{N}{\sqrt{4\pi N\log(N)}}}\right)$ for $\delta > 0$ and $\phi(\delta) = \frac{e^{-\delta^2/2}}{\sqrt{2\pi}}$. This new bound is **consistently positive and offers a higher probability**, especially when the number of arms $N$ is large.
>
> We theoretically prove the tightened bound in the revised Appendix B, draw the function of $p(N) = \left(1 - e^{-2(\log(N)-1 )^2}\right)\left(1-e^{-\frac{N}{\sqrt{4\pi N\log(N)}}}\right)$ and $f(\delta) = 1-\frac{\delta}{1+ \delta^2}\phi(\delta)$ in Figure 10 of Appendix E, and discussed its trend in the **Appendix E**.
>
> Specifically, the minimum of $f(\delta)$ is 0.852 in the revised probability bound. As for $p(N)$, typically, $N$ is large in practical scenarios due to its combinatorial nature, even for a relatively small number of clients, and E-TS is designed to manage this extensive exploration space. For instance, in Figure 4, the number of arms $N$ is at least 120, implying $p(N) \geq 0.756$. For smaller $N $ values, such as $N = 21$ with $p(N) = 0.52$ in the FashionMNIST attack (Figure 1 (a)), the lower probability bound may seem modest, yet the empirical performance demonstrates rapid convergence compared to plain TS. Overall, our E-TS algorithm consistently shows faster convergence than TS in practice, and we can theoretically guarantee a high probability bound for large $N$.
>
> [5] Andriushchenko et al. (2020) Square Attack: a query-efficient black-box adversarial attack via random search
>
> [6] Chen et al. (2023) Rethinking Model Ensemble in Transfer-based Adversarial Attacks

---

> ### Author Response · Authors · 2023-11-19
> **Global response (Part 3/3)**
>
> 5. The fifth concern is the **advantage** of transforming the corruption pattern selection (CPS) problem into a multi-armed bandits (MAB) problem rather than a combinatorial MAB (CMAB) problem, proposed by reviewers HdL1 and qKnn.
>
> To clarify, we initially considered CMAB for the corruption pattern selection issue but ultimately chose MAB for several reasons:
>
> **Clear Definition of Rewards:** In a CMAB framework, defining the reward for each base arm is challenging, especially under varying corruption constraints. For instance, using the average ASR (Attack Success Rate) to represent a base arm's reward can result in inconsistencies across different constraints. MAB, on the other hand, allows for a straightforward definition of each arm's reward as the ASR when that particular arm is used.
>
> **Simplified Assumptions:** CMAB typically assumes the rewards of base arms to be independent and identically distributed (i.i.d.), which does not hold in our context due to the influence of corruption constraints. Moreover, common CMAB models [7] [8] assume that the super arm's reward is a simple aggregate (average or sum) of the base arms' rewards, which does not align with our attack model. Although some CMAB variants [9] [10] consider nonlinear relationships between super and base arms, they still rely on observable rewards for base arms or presume i.i.d. distributions, which are not applicable in our case.
>
> **Comparative Performance and Regret Bound:** The regret bound in most CMAB works is $\mathcal{O}(\log(T))$ [7] [8] [9] [10], similar to MAB solutions. However, our MAB approach improves the regret of non-competitive arms from $\mathcal{O}(\log(T))$ to
> $\mathcal{O}(\log(1))$, showing a significant advantage. Additionally, our initial trials with a standard CMAB algorithm [7] yielded poorer results compared to a plain Thompson Sampling (TS) approach in MAB.
>
> Therefore, considering these aspects, MAB was chosen for its clearer reward structure, less restrictive assumptions, and superior performance in our specific scenario.
>
>
> 6. The last concern, raised by reviewers GNem and XBEx, revolves around the clarity of the **parameter settings** in our ablation study.
>
> Regarding the default parameter settings used in our ablation study, they are as follows: $\beta = 0.3, B^t = 128$, and $C = 2$. In the ablation study, we varied test parameters while maintaining the stability of other settings. The comprehensive parameter settings are now detailed in the revised draft (7.1 Setup) and **Table 1 of Appendix C.1**.
>
> [7] Wang et al. (2018) Thompson Sampling for Combinatorial Semi-Bandits
>
> [8] Combe et al. (2015) Combinatorial Bandits Revisited
>
> [9] Chen et al. (2016) Combinatorial Multi-Armed Bandit with General Reward Functions
>
> [10] Perrault et al. (2020) Statistical Efficiency of Thompson Sampling for Combinatorial Semi-Bandits
>
>
>
> Here, we list all the modifications of the draft and Appendix. The modified parts have been noted in **red color** in the revised draft and Appendix.
>
>
> |         Modified Part        |     Description|
> | :---        |    :----------   |
> | Algorithm 1  | Include Algorithm 2 in line 12       |
> | Section 6 Regret analysis     | Modify the probability bound of Lemma 1, 2 and Theorem 1      |
> | Section 7.1 setup  |    Add the settings of ablation study   |
> | Appendix B| Correct Fact 1, tighten the probability bound of Lemma 3, and modify other Lemmas accordingly|
> |Appendix C.3| Test the untargeted attack performance on CIFAT-10 in perturbation budget $\beta = 0.08$|
> | Appendix C.4| The study of the optimal choice of $t_0$|
> | Appendix C.5| The study on dynamics of arm selection and empirical competitive set in TS and E-TS|
> | Appendix D | Minimum query budget and corruption channels to achieve 50\% ASR |
> | Appendix E | Discussion on the probability of the regret bound |
> | Appendix F| Discussion on the large exploration spaces |
> | Appendix G| Numerical experiments among E-TS, TS, and nonlinear sequential elimination methods|

---

### Meta-Review · Area_Chair_5N9o · 2023-12-06

**Metareview:**

The paper considered adversarial attacks on Vertical Federated Learning (VFL) where the adversary is able to adaptively corrupt a subset of clients. The authors showed that corruption pattern selection problem is equivalent to a multi-armed bandit (MAB) problem, and proposed Thompson sampling with Empirical maximum reward (E-TS) algorithm to efficiently identify the best clients to corrupt.

There are extensive discussion between author-reviewer and reviewer-AC. After revision, all reviewers appreciate the novel setting, sound theoretical analysis and empirical results. One major concern is about the regret bound in Theorem 1. While the reviewers and AC does not have concerns about correctness, the main question is whether it is appropriate to present the result as a high probability conditional regret bound. As suggested by Reviewer HdL1, the authors should discuss the rate of an unconditional regret bound in the final version.

**Justification For Why Not Higher Score:**

There is a concern regarding the presentation of Theorem 1.

**Justification For Why Not Lower Score:**

The reviewers recognize the novel setting, sound theoretical analysis and empirical results.

---

### Decision · Program_Chairs · 2024-01-16

Accept (poster)